# Kernel Quadrature with Randomly Pivoted Cholesky

**Ethan N. Epperly and Elvira Moreno**[*]
Department of Computing and Mathematical Sciences
California Institute of Technology
Pasadena, CA 91125
{eepperly,emoreno2}@caltech.edu

## Abstract

This paper presents new quadrature rules for functions in a reproducing kernel Hilbert space using nodes drawn by a sampling algorithm known as randomly pivoted Cholesky. The resulting computational procedure compares favorably to previous kernel quadrature methods, which either achieve low accuracy or require solving a computationally challenging sampling problem. Theoretical and numerical results show that randomly pivoted Cholesky is fast and achieves comparable quadrature error rates to more computationally expensive quadrature schemes based on continuous volume sampling, thinning, and recombination. Randomly pivoted Cholesky is easily adapted to complicated geometries with arbitrary kernels, unlocking new potential for kernel quadrature.

## 1 Introduction

Quadrature is one of the fundamental problems in computational mathematics, with applications in Bayesian statistics [35], probabilistic ODE solvers [27], reinforcement learning [32], and model-based machine learning [30]. The task is to approximate an integral of a function $f$ by the weighted sum of $f$'s values at judiciously chosen quadrature points $s_1, \ldots, s_n$:

$$\int_{\mathsf{X}} f(x)g(x)\,\mathrm{d}\mu(x) \approx \sum_{i=1}^{n} w_i f(s_i). \tag{1}$$

Here, and throughout, $\mathsf{X}$ denotes a topological space equipped with a Borel measure $\mu$, and $g \in \mathsf{L}^2(\mu)$ denotes a square-integrable function. The goal of kernel quadrature is to select quadrature weights $w_1, \ldots, w_n \in \mathbb{R}$ and nodes $s_1, \ldots, s_n \in \mathsf{X}$ which minimize the error in the approximation (1) for all $f$ drawn from a reproducing kernel Hilbert space (RKHS) $\mathcal{H}$ of candidate functions.

The ideal kernel quadrature scheme would satisfy three properties:

1. *Spectral accuracy.* The error of the approximation (1) decreases at a rate governed by the eigenvalues of the reproducing kernel $k$ of $\mathcal{H}$, with rapidly decaying eigenvalues guaranteeing rapidly decaying quadrature error.

2. *Efficiency.* The nodes $s_1, \ldots, s_n$ and weights $w_1, \ldots, w_n$ can be computed by an algorithm which is efficient in both theory and practice.

---

[*]ENE acknowledges support by NSF FRG 1952777, Carver Mead New Horizons Fund, and the U.S. Department of Energy, Office of Science, Office of Advanced Scientific Computing Research, Department of Energy Computational Science Graduate Fellowship under Award Number DE-SC0021110. EM was supported in part by Air Force Office of Scientific Research grant FA9550-22-1-0225.

The first two goals may be more easily achieved if one has access to a Mercer decomposition

$$k(x,y) = \sum_{i=1}^{\infty} \lambda_i e_i(x) e_i(y), \tag{2}$$

where $e_1, e_2, \ldots$ form an orthonormal basis of $\mathsf{L}^2(\mu)$ and the eigenvalues $\lambda_1 \geq \lambda_2 \geq \cdots$ are decreasing. Fortunately, the Mercer decomposition is known analytically for many RKHS's $\mathcal{H}$ on simple sets $\mathsf{X}$ such as boxes $[0,1]^d$. However, such a decomposition is hard to compute for general kernels $k$ and domains $\mathsf{X}$, leading to the third of our desiderata:

3. *Mercer-free.* The quadrature scheme can be efficiently implemented without access to an explicit Mercer decomposition (2).

Despite significant progress in the probabilistic and kernel quadrature literature (see, e.g., [3–8, 17, 18, 21–23, 26, 41]), the search for a kernel quadrature scheme meeting all three criteria remains ongoing.

**Contributions.**    We present a new kernel quadrature method based on the *randomly pivoted Cholesky* (RPCHOLESKY) sampling algorithm that achieves all three of the goals. Our main contributions are

1. Generalizing RPCHOLESKY sampling (introduced in [10] for finite kernel matrices) to the continuum setting and demonstrating its effectiveness for kernel quadrature.

2. Establishing theoretical results (see theorem 1) which show that RPCHOLESKY kernel quadrature achieves near-optimal quadrature error rates.

3. Developing efficient rejection sampling implementations (see algorithms 2 and 4) of RP-CHOLESKY in the continuous setting, allowing kernel quadrature to be applied to general spaces $\mathsf{X}$, measures $\mu$, and kernels $k$ with ease.

The remainder of this introduction will sketch our proposal for RPCHOLESKY kernel quadrature. A comparison with existing kernel quadrature approaches appears in section 2.

**Randomly pivoted Cholesky.**    Let $\mathcal{H}$ be an RKHS with a kernel $k$ that is integrable on the diagonal

$$\int_{\mathsf{X}} k(x,x) \, \mathrm{d}\mu(x) < +\infty. \tag{3}$$

RPCHOLESKY uses the kernel diagonal $k(x,x)$ as a *sampling distribution* to pick quadrature nodes. The first node $s_1$ is chosen to be a sample from the distribution $k(x,x) \, \mathrm{d}\mu(x)$, properly normalized:

$$s_1 \sim \frac{k(x,x) \, \mathrm{d}\mu(x)}{\int_{\mathsf{X}} k(x,x) \, \mathrm{d}\mu(x)}.$$

Having selected $s_1$, we remove its influence on the kernel, updating the entire kernel function:

$$k'(x,y) = k(x,y) - \frac{k(x,s_1)k(s_1,y)}{k(s_1,s_1)}. \tag{4}$$

Linear algebraically, the update (4) can be interpreted as Gaussian elimination, eliminating "row" and "column" $s_1$ from the "infinite matrix" $k$. Probabilistically, if we interpret $k$ as the covariance function of a Gaussian process, the update (4) represents conditioning on the value of the process at $s_1$. We use the updated kernel $k'$ to select the next quadrature node:

$$s_2 \sim \frac{k'(x,x) \, \mathrm{d}\mu(x)}{\int_{\mathsf{X}} k'(x,x) \, \mathrm{d}\mu(x)},$$

whose influence is then subtracted off $k'$ as in (4). RPCHOLESKY continues along these lines until $n$ nodes have been selected. The resulting algorithm is shown in algorithm 1. Having chosen the nodes $s_1, \ldots, s_n$, our choice of weights $w_1, \ldots, w_n$ is standard and is discussed in section 3.2.

RPCHOLESKY sampling is more flexible than many kernel quadrature methods, easily adapting to general spaces $\mathsf{X}$, measures $\mu$, and kernels $k$. To demonstrate this flexibility, we apply RPCHOLESKY to the region $\mathsf{X}$ in fig. 1a equipped with the Matérn $5/2$-kernel with bandwidth 2 and the measure

$$\mathrm{d}\mu(x,y) = (x^2 + y^2) \, \mathrm{d}x \, \mathrm{d}y.$$

**Algorithm 1** RPCHOLESKY: unoptimized implementation

---

**Input:** Kernel $k : \mathsf{X} \times \mathsf{X} \to \mathbb{R}$ and number of quadrature points $n$
**Output:** Quadrature points $s_1, \ldots, s_n$
 1: **for** $i = 1, 2, \ldots, n$ **do**
 2:     Sample $s_i$ from the probability measure $k(x, x) \, \mathrm{d}\mu(x) / \int_{\mathsf{X}} k(x, x) \, \mathrm{d}\mu(x)$
 3:     Update kernel $k \leftarrow k - k(\cdot, s_i) k(s_i, s_i)^{-1} k(s_i, \cdot)$
 4: **end for**

---

A set of $n = 20$ quadrature nodes produced by RPCHOLESKY sampling (using algorithm 2) is shown in fig. 1a. The cyan–pink shading shows the diagonal of the kernel after updating for the selected nodes. We see that near the selected nodes, the updated kernel is very small, meaning that future steps of the algorithm will avoid choosing nodes in those regions. Nodes far from any currently selected nodes have a much larger kernel value, making them more likely to be chosen in future RPCHOLESKY iterations.

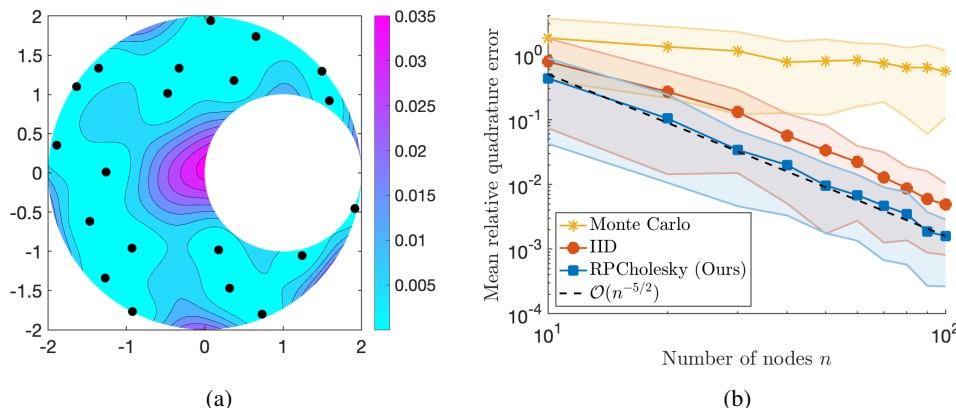

(a)                   (b)

Figure 1: **RPCHOLESKY kernel quadrature on oddly shaped region.** *Left:* Black dots show 20 points selected by RPCHOLESKY on a crescent-shaped region. Shading shows kernel values $k((x, y), (x, y))$ after removing influence of selected points. *Right:* Mean relative error for RPCHOLESKY kernel quadrature, iid kernel quadrature, and Monte Carlo quadrature for $f(x, y) = \sin(x) \exp(y)$ with 100 trials. Shaded regions show 10%/90% quantiles.

The quadrature error for $f(x, y) = \sin(x) \exp(y)$, $g \equiv 1$, and different numbers $n$ of quadrature nodes for RPCHOLESKY kernel quadrature, kernel quadrature with nodes drawn iid from $\mu/\mu(\mathsf{X})$, and Monte Carlo quadrature are shown in fig. 1b. Other spectrally accurate kernel quadrature methods would be difficult to implement in this setting because they would require an explicit Mercer decomposition (projection DPPs and leverage scores) or an expensive sampling procedure (continuous volume sampling). A comparison of RPCHOLESKY with more kernel quadrature methods on a benchmark problem is provided in fig. 2.

## 2 Related work

Here, we overview past work on kernel quadrature and discuss the history of RPCHOLESKY sampling.

### 2.1 Kernel quadrature

The goal of kernel quadrature is to provide a systematic means for designing quadrature rules on RKHS's. Relevant points of comparison are Monte Carlo [36] and quasi-Monte Carlo [14] methods, which have $\mathcal{O}(1/\sqrt{n})$ and $\mathcal{O}(\mathrm{polylog}(n)/n)$ convergence rates, respectively.

The literature on probabilistic and kernel approaches to quadrature is vast, so any short summary is necessarily incomplete. Here, we briefly summarize some of the most prominent kernel quadrature

methods, highlighting limitations that we address with our RPCHOLESKY approach. We refer the reader to [22, Tab. 1] for a helpful comparison of many of the below-discussed methods.

**Herding.** Kernel herding schemes [4, 11, 26, 41] iteratively select quadrature nodes using a greedy approach. These methods face two limitations. First, they require the solution of a (typically) nonconvex global optimization problem at every step, which may be computationally costly. Second, these methods exhibit slow $\mathcal{O}(1/n)$ quadrature error rates [11, Prop. 4] (or even slower [16]). Optimally weighted herding is known as sequential Bayesian quadrature [26].

**Thinning.** Thinning methods [17, 18, 38] try to select $n$ good quadrature nodes from a larger set of $N \gg n$ candidate nodes drawn from a simple distribution. While these methods have other desirable theoretical properties, they do not benefit from spectral accuracy.

**Leverage scores and projection DPPs.** With optimal weights (section 3.2), quadrature with nodes sampled using a projection determinal point process (DPP) [7] (see also [5, 6, 21]) or iid from the (ridge) leverage score distribution [3] achieve spectral accuracy. However, known efficient sampling algorithms require access to the Mercer decomposition (2), limiting the applicability of these schemes to simple spaces X, measures $\mu$, and kernels $k$, where the decomposition is known analytically.

**Continuous volume sampling.** Continuous volume sampling is the continuous analog of $n$-DPP sampling [28], providing quadrature nodes that achieve spectral accuracy [8, Prop. 5]. Unfortunately, continuous volume sampling is computationally challenging. In the finite setting, the best-known algorithm [9] for exact $n$-DPP sampling requires a costly $\mathcal{O}(|\mathsf{X}| \cdot n^{6.5} + n^{9.5})$ operations. Inexact samplers based on Markov chain Monte Carlo (MCMC) (e.g., [1, 2, 34]) may be more competitive, but the best-known samplers in the continuous setting still require an expensive $\mathcal{O}(n^5 \log n)$ MCMC steps [34, Thms. 1.3–1.4]. RPCHOLESKY sampling achieves similar theoretical guarantees to continuous volume sampling (see theorem 1) and can be efficiently and exactly sampled (algorithm 2).

**Recombination and convex weights.** The paper [22] (see also [23]) proposes two ideas for kernel quadrature when $\mu$ is a probability measure and $g \equiv 1$. First, they suggest using a recombination algorithm (e.g., [29]) to subselect good quadratures from $N \gg n$ candidate nodes iid sampled from $\mu$. All of the variants of their method either fail to achieve spectral accuracy or require an explicit Mercer decomposition [22, Tab. 1]. Second, they propose choosing weights $(w_1, \ldots, w_n)$ that are convex combination coefficients. This choice makes the quadrature scheme robust against misspecification of the RKHS $\mathcal{H} \not\ni f$, among other benefits. It may be worth investigating a combination of RPCHOLESKY quadrature nodes with convex weights in future work.

## 2.2 Randomly pivoted Cholesky

RPCHOLESKY was proposed, implemented, and analyzed in [10] for the task of approximating an $M \times M$ positive-semidefinite matrix $\boldsymbol{A}$. The algorithm is algebraically equivalent to applying an earlier algorithm known as *adaptive sampling* [12, 13] to $\boldsymbol{A}^{1/2}$. Despite their similarities, RPC-HOLESKY and adaptive sampling are different algorithms: To produce a rank-$n$ approximation to an $M \times M$ matrix, RPCHOLESKY requires $\mathcal{O}(n^2 M)$ operations, while adaptive sampling requires a much larger $\mathcal{O}(n M^2)$ operations. See [10, §4.1] for further discussion on RPCHOLESKY's history. In this paper, we introduce a continuous extension of RPCHOLESKY and analyze its effectiveness for kernel quadrature.

# 3 Theoretical results

In this section, we prove our main theoretical result for RPCHOLESKY kernel quadrature. We first establish the mathematical setting (section 3.1) and introduce kernel quadrature (section 3.2). Then, we present our main theorem (section 3.3) and discuss its proof (section 3.4).

## 3.1 Mathematical setting and notation

Let $\mu$ be a Borel measure supported on a topological space X and let $\mathcal{H}$ be a RKHS on X with continuous kernel $k : \mathsf{X} \times \mathsf{X} \to \mathbb{R}$. We assume that $x \mapsto k(x, x)$ is integrable (3) and that $\mathcal{H}$

is dense in $L^2(\mu)$. These assumptions imply that $\mathcal{H}$ is compactly embedded in $L^2(\mu)$, the Mercer decomposition (2) converges pointwise, and the Mercer eigenfunctions form an orthonormal basis of $L^2(\mu)$ and an orthogonal basis of $\mathcal{H}$ [39, Thm. 3.1].

Define the integral operator

$$T f(x) = \int_{\mathsf{X}} k(x, y) f(y) \, \mathrm{d}\mu(y). \tag{5}$$

Viewed as an operator $T : L^2(\mu) \to L^2(\mu)$, $T$ is self-adjoint, positive semidefinite, and trace-class.

One final piece of notation: For a function $\delta : \mathsf{X} \to \mathbb{R}$ and an ordered (multi)set $\mathsf{S} = \{s_1, \ldots, s_n\}$, we let $\delta(\mathsf{S})$ be the column vector with $i$th entry $\delta(s_i)$. Similarly, for a bivariate function $h : \mathsf{X} \times \mathsf{X} \to \mathbb{R}$, $h(\cdot, \mathsf{S})$ (resp. $h(\mathsf{S}, \cdot)$) denotes the row (resp. column) vector-valued function with $i$th entry $h(\cdot, s_i)$ (resp. $h(s_i, \cdot)$), and $h(\mathsf{S}, \mathsf{S})$ denotes the matrix with $ij$ entry $h(s_i, s_j)$.

## 3.2 Kernel quadrature

Following earlier works (e.g., [3, 7]), let us describe the kernel quadrature problem more precisely. Given a function $g \in L^2(\mu)$, we seek quadrature weights $\boldsymbol{w} = (w_1, \ldots, w_n) \in \mathbb{R}^n$ and nodes $\mathsf{S} = \{s_1, \ldots, s_n\} \subseteq \mathsf{X}$ that minimize the maximum quadrature error over all $f \in \mathcal{H}$ with $\|f\|_{\mathcal{H}} \leq 1$:

$$\mathrm{Err}(\mathsf{S}, \boldsymbol{w}; g) \coloneqq \max_{\|f\|_{\mathcal{H}} \leq 1} \left| \int_{\mathsf{X}} f(x) g(x) \, \mathrm{d}\mu(x) - \sum_{i=1}^{n} w_i f(s_i) \right|. \tag{6}$$

A short derivation ([3, Eq. (7)] or appendix C) yields the simple formula

$$\mathrm{Err}(\mathsf{S}, \boldsymbol{w}; g) = \left\| T g - \sum_{i=1}^{n} w_i k(\cdot, s_i) \right\|_{\mathcal{H}}. \tag{7}$$

This equation reveals that the quadrature rule minimizing $\mathrm{Err}(\mathsf{S}, \boldsymbol{w}; g)$ is the least-squares approximation of $T g \in \mathcal{H}$ by a linear combination of kernel function evaluations $\sum_{i=1}^{n} w_i k(\cdot, s_i)$.

If we fix the quadrature nodes $\mathsf{S} = \{s_1, \ldots, s_n\}$, the optimal weights $\boldsymbol{w}_\star = (w_{\star 1}, \ldots, w_{\star n}) \in \mathbb{R}^n$ minimizing $\mathrm{Err}(\mathsf{S}, \boldsymbol{w}; g)$ are the solution of the linear system of equations

$$k(\mathsf{S}, \mathsf{S}) \boldsymbol{w}_\star = T g(\mathsf{S}). \tag{8}$$

We use (8) to select the weights throughout this paper, and denote the error with optimal weights by

$$\mathrm{Err}(\mathsf{S}; g) \coloneqq \mathrm{Err}(\mathsf{S}, \boldsymbol{w}_\star; g). \tag{9}$$

## 3.3 Error bounds for randomly pivoted Cholesky kernel quadrature

Our main result for RPCHOLESKY kernel quadrature is as follows:

**Theorem 1.** *Let* $\mathsf{S} = \{s_1, \ldots, s_n\}$ *be generated by the* RPCHOLESKY *sampling algorithm. For any function* $g \in L^2(\mu)$, *nonnegative integer* $r \geq 0$, *and real number* $\delta > 0$, *we have*

$$\mathbb{E} \, \mathrm{Err}^2(\mathsf{S}; g) \leq \delta \cdot \|g\|_{L^2(\mu)}^2 \cdot \sum_{i=r+1}^{\infty} \lambda_i$$

*provided*

$$n \geq r \log \left( \frac{2\lambda_1}{\delta \sum_{i=r+1}^{\infty} \lambda_i} \right) + \frac{2}{\delta}. \tag{10}$$

To illustrate this result, consider an RKHS with eigenvalues $\lambda_i = \mathcal{O}(i^{-2s})$ for $s > 1/2$. An example is the periodic Sobolev space $H_{\mathrm{per}}^s([0, 1])$ (see section 5). By setting $\delta = 1/r$, we see that

$$\mathbb{E} \, \mathrm{Err}^2(\mathsf{S}; g) \leq \mathcal{O}(r^{-2s}) \|g\|_{L^2(\mu)}^2 \quad \text{for RPCHOLESKY with } n = \Omega(r \log r).$$

The optimal scheme requires $n = \Theta(r)$ nodes to achieve this bound [8, Prop. 5 & §2.5]. Thus, to achieve an error of $\mathcal{O}(r^{-2s})$, **RPCHOLESKY requires just logarithmically more nodes than**

**the optimal quadrature scheme for such spaces.** This compares favorably to continuous volume sampling, which achieves the slightly better optimal rate but is much more difficult to sample.

Having established that RPCHOLESKY achieves nearly optimal error rates for interesting RKHS's, we make some general comments on theorem 1. First, observe that the error depends on the sum $\sum_{i=r+1}^{\infty} \lambda_i$ of the tail eigenvalues. This tail sum is characteristic of spectrally accurate kernel quadrature schemes [7, 8]. The more distinctive feature in (10) is the logarithmic factor. Fortunately, for double precision computation, the achieveable accuracy is bounded by the machine precision $2^{-52}$, so this logarithmic factor is effectively a modest constant:

$$\log \left( \frac{2\lambda_1}{\delta \sum_{i=r+1}^{\infty}} \right) \lesssim \log(2^{52}) < 37.$$

### 3.4 Connection to Nyström approximation and idea of proof

We briefly outline the proof of theorem 1. Following previous works (e.g., [3, 23]), we first utilize the connection between kernel quadrature and the Nyström approximation [31, 42] of the kernel $k$.

**Definition 2.** For nodes $\mathsf{S} = \{s_1, \ldots, s_n\} \subseteq \mathsf{X}$, the *Nyström approximation* to the kernel $k$ is

$$k_\mathsf{S}(x, y) = k(x, \mathsf{S}) \, k(\mathsf{S}, \mathsf{S})^\dagger \, k(\mathsf{S}, y). \tag{11}$$

Here, $^\dagger$ denotes the Moore–Penrose pseudoinverse. The Nyström approximate integral operator is

$$T_\mathsf{S} f := \int_\mathsf{X} k_\mathsf{S}(\cdot, x) f(x) \, \mathrm{d}\mu(x).$$

This definition leads to a formula for the quadrature rule $\sum_{i=1}^n w_i k(\cdot, s_i)$ with optimal weights (8).

**Proposition 3.** *Fix nodes* $\mathsf{S} = \{s_1, \ldots, s_n\}$. *With the optimal weights* $\boldsymbol{w}_\star \in \mathbb{R}^n$ (8), *we have*

$$\sum_{i=1}^n w_{i\star} k(\cdot, s_i) = T_\mathsf{S} g. \tag{12}$$

*Consequently,*

$$\mathrm{Err}^2(\mathsf{S}; g) = \|(T - T_\mathsf{S})g\|_\mathcal{H}^2 \leq \langle g, (T - T_\mathsf{S})g \rangle_{\mathsf{L}^2(\mu)}.$$

To finish the proof of theorem 1, we develop and solve a recurrence for an upper bound on the largest eigenvalue of $\mathbb{E}[T - T_\mathsf{S}]$. See appendix A for details and for the proof of proposition 3.

## 4 Efficient randomly pivoted Cholesky by rejection sampling

To efficiently perform RPCHOLESKY sampling in the continuous setting, we can use *rejection sampling*. (A similar rejection sampling idea is used in the MCMC continuous volume sampler of [34].) Assume for simplicity that the measure $\mu$ is normalized so that

$$\int_\mathsf{X} k(x, x) \, \mathrm{d}\mu(x) = \mathrm{Tr}\, T = 1.$$

We assume two forms of access to the kernel $k$:

1. **Entry evaluations.** For any $x, y \in \mathsf{X}$, we can evaluate $k(x, y)$.

2. **Sampling the diagonal.** We can produce samples from the measure $k(x, x) \, \mathrm{d}\mu(x)$.

To sample from the RPCHOLESKY distribution, we use $k(x, x) \, \mathrm{d}\mu(x)$ as a proposal distribution and accept proposal $s$ with probability

$$1 - \frac{k_\mathsf{S}(s, s)}{k(s, s)}.$$

(Recall $k_\mathsf{S}$ from (11).) The resulting algorithm for RPCHOLESKY sampling is shown in algorithm 2.

---

**Algorithm 2** RPCHOLESKY with rejection sampling

---

**Input:** Kernel $k : \mathsf{X} \times \mathsf{X} \to \mathbb{R}$ and number of quadrature points $n$
**Output:** Quadrature points $s_1, \ldots, s_n$
 1: Initialize $\boldsymbol{L} \leftarrow \boldsymbol{0}_{n \times n}$, $i \leftarrow 1$, $\mathsf{S} \leftarrow \emptyset$      $\triangleright$ $\boldsymbol{L}$ stores a Cholesky decomposition of $k(\mathsf{S}, \mathsf{S})$
 2: **while** $i \leq n$ **do**
 3:    Sample $s_i$ from the probability measure $k(x,x)\,\mathrm{d}\mu(x)$
 4:    $(d, \boldsymbol{c}) \leftarrow \textsc{ResidualKernel}(s_i, \mathsf{S}, \boldsymbol{L}, k)$     $\triangleright$ Helper subroutine algorithm 3
 5:    Draw a uniform random variable $U \sim \textsc{Unif}[0,1]$
 6:    **if** $U < d/k(s_i, s_i)$ **then**       $\triangleright$ Accept with probability $d/k(s_i, s_i)$
 7:      Induct $s_i$ into the selected set: $\mathsf{S} \leftarrow \mathsf{S} \cup \{s_i\}$, $i \leftarrow i + 1$
 8:      $\boldsymbol{L}(i, 1:i) \leftarrow \begin{bmatrix} \boldsymbol{c}^\top & \sqrt{d} \end{bmatrix}$
 9:    **end if**
10: **end while**

---

---

**Algorithm 3** Helper subroutine to evaluate residual kernel

---

**Input:** Point $s \in \mathsf{X}$, set $\mathsf{S} \subseteq \mathsf{X}$, Cholesky factor $\boldsymbol{L}$ of $k(\mathsf{S}, \mathsf{S})$, and kernel $k$
**Output:** $d = k(s,s) - k_\mathsf{S}(s,s)$ and $\boldsymbol{c} = \boldsymbol{L}^{-1} k(\mathsf{S}, s)$
 1: **procedure** RESIDUALKERNEL($s$,$\mathsf{S}$,$\boldsymbol{L}$,$k$)
 2:    $\boldsymbol{c} \leftarrow \boldsymbol{L}^{-1} k(\mathsf{S}, s)$
 3:    $d \leftarrow k(s,s) - \|\boldsymbol{c}\|^2$           $\triangleright$ $d = k(s,s) - k_\mathsf{S}(s,s)$
 4:    **return** $(d, \boldsymbol{c})$
 5: **end procedure**

---

**Theorem 4.** *Algorithm 2 produces exact* RPCHOLESKY *samples. Let $\eta_i$ denote the trace-error of the best rank-$i$ approximation to $T$:*

$$\eta_i = \sum_{j=i+1}^{\infty} \lambda_i.$$

*The expected runtime of algorithm 2 is at most $\mathcal{O}(\sum_{i=1}^n i^2/\eta_i) \leq \mathcal{O}(n^3/\eta_{n-1})$.*

This result demonstrates that algorithm 2 suffers from the *curse of smoothness*: The faster the eigenvalues $\lambda_1, \lambda_2, \ldots$ decrease, the smaller $\eta_{n-1}$ will be and, consequently, the slower algorithm 2 will be in expectation. While this curse is an unfortunate limitation, it is also a common one. The curse of smoothness affects all known Mercer-free, spectrally accurate kernel quadrature schemes. In fact, the situation is worse for other algorithms. The CVS sampler of [34], for example, requires as many as $\mathcal{O}(n^5 \log n)$ MCMC steps, each of which has cost $\mathcal{O}(n^2/\eta_{n-1})$. According to current analysis, algorithm 2 is $\mathcal{O}(n^4 \log n)$-times faster than the CVS sampler of [34].

Notwithstanding the curse of smoothness, algorithm 2 is useful in practice. The algorithm works under minimal assumptions and can reach useful accuracies $\eta = 10^{-5}$ while sampling $n = 1000$ nodes on a laptop. Algorithm 2 may also be interesting for RPCHOLESKY sampling for a large finite set $|\mathsf{X}| \geq 10^9$, since its runtime has no explicit dependence on the size of the space $\mathsf{X}$.

To achieve higher accuracies and blunt the curse of smoothness, we can improve algorithm 2 with optimization. Indeed, the optimal acceptance probability for algorithm 2 would be

$$p(s_i; \alpha) = \frac{1}{\alpha} \cdot \frac{k(s_i, s_i) - k_\mathsf{S}(s_i, s_i)}{k(s_i, s_i)} \quad \text{where} \quad \alpha = \max_{x \in \mathsf{X}} \frac{k(x,x) - k_\mathsf{S}(x,x)}{k(x,x)}.$$

This suggests the following scheme: Initialize with $\alpha = 1$ and run the algorithm with acceptance probability $p(s_i; \alpha)$. If we perform many loop iterations without an acceptance, we then recompute the optimal $\alpha$ by solving an optimization problem. The resulting procedure is shown in algorithm 4.

If the optimization problem for $\alpha$ is solved to global optimality, then algorithm 4 produces exact RPCHOLESKY samples. The downside of algorithm 4 is the need to solve a (typically) nonconvex global optimization problem to compute the optimal $\alpha$. Fortunately, in our experiments (section 5), only a small number of optimization problems ($\leq 10$) are needed to produce a sample of $n = 1000$ nodes. In the setting where the optimization problem is tractable, the speedup of algorithm 4 can

**Algorithm 4** RPCHOLESKY with optimized rejection sampling

---

**Input:** Kernel $k : \mathsf{X} \times \mathsf{X} \to \mathbb{R}$ and number of quadrature points $n$
**Output:** Quadrature points $s_1, \dots, s_n$
1: Initialize $\boldsymbol{L} \leftarrow \boldsymbol{0}_{n \times n}$, $i \leftarrow 1$, $\mathsf{S} \leftarrow \emptyset$, $\texttt{trials} \leftarrow 0$, $\alpha \leftarrow 1$
2: **while** $i \le n$ **do**
3: $\quad$ $\texttt{trials} \leftarrow \texttt{trials} + 1$
4: $\quad$ Sample $s_i$ from the probability measure $k(x, x)\, \mathrm{d}\mu(x)$
5: $\quad$ $(d, \boldsymbol{c}) \leftarrow \textsc{ResidualKernel}(s_i, \mathsf{S}, \boldsymbol{L}, k)$ $\qquad\qquad$ ▷ Helper subroutine algorithm 3
6: $\quad$ Draw a uniform random variable $U \sim \textsc{Unif}[0, 1]$
7: $\quad$ **if** $U < (1/\alpha) \cdot d/k(s_i, s_i)$ **then** $\qquad\qquad$ ▷ Accept with probability $d/k(s_i, s_i)$
8: $\quad\quad$ $\mathsf{S} \leftarrow \mathsf{S} \cup \{s_i\}$, $i \leftarrow i + 1$, $\texttt{trials} \leftarrow 0$
9: $\quad\quad$ $\boldsymbol{L}(i, 1 : i) \leftarrow \begin{bmatrix} \boldsymbol{c}^\top & \sqrt{d} \end{bmatrix}$
10: $\quad$ **end if**
11: $\quad$ **if** $\texttt{trials} \ge \texttt{TRIALS\_MAX}$ **then** $\qquad\qquad$ ▷ We use $\texttt{TRIALS\_MAX} \in [25, 1000]$
12: $\quad\quad$ $\alpha \leftarrow \max_{x \in \mathsf{X}} \textsc{ResidualKernel}(x, \mathsf{S}, \boldsymbol{L}, k)/k(x, x)$
13: $\quad\quad$ $\texttt{trials} \leftarrow 0$
14: $\quad$ **end if**
15: **end while**

---

be immense. To produce $n = 200$ RPCHOLESKY samples for the space $[0, 1]^3$ with the kernel (13), algorithm 4 requires just $0.19$ seconds compared to $7.47$ seconds for algorithm 2.

## 5 Comparison of methods on a benchmark example

Having demonstrated the versatility of RPCHOLESKY for general spaces $\mathsf{X}$, measures $\mu$, and kernels $k$ in fig. 1, we now present a benchmark example from the kernel quadrature literature to compare RPCHOLESKY with other methods. Consider the periodic Sobolev space $H^s_{\mathrm{per}}([0, 1])$ with kernel

$$k(x, y) = 1 + 2 \sum_{m=1}^{\infty} m^{-2s} \cos(2\pi m(x - y)) = 1 + \frac{(-1)^{s-1}(2\pi)^{2s}}{(2s)!} B_{2s}(\{x - y\}),$$

where $B_{2s}$ is the $(2s)$th Bernoulli polynomial and $\{\cdot\}$ reports the fractional part of a real number [3, p. 7]. We also consider $[0, 1]^3$ equipped with the product kernel

$$k^{\otimes 3}((x_1, x_2, x_3), (y_1, y_2, y_3)) = k(x_1, y_1)k(x_2, y_2)k(x_3, y_3). \tag{13}$$

We quantify the performance of nodes $\mathsf{S}$ and weights $\boldsymbol{w}$ using $\mathrm{Err}(\mathsf{S}, \boldsymbol{w}; g)$, which can be computed in closed form [22, Eq. (14)]. We set $\mu := \textsc{Unif}[0, 1]^d$ and $g \equiv 1$.

We compare the following schemes:

- *Monte Carlo, IID kernel quadrature.* Nodes $s_1, \dots, s_n \overset{\text{iid}}{\sim} \mu$ with uniform weights $w_i = 1/n$ (Monte Carlo) and optimal weights (8) (IID).

- *Thinning.* Nodes $s_1, \dots, s_n$ thinned from $n^2$ iid samples from $\mu = \textsc{Unif}[0, 1]$ using kernel thinning [17, 18] with the COMPRESS++ algorithm [38] with optimal weights (8).

- *Continuous volume sampling (CVS).* Nodes $s_1, \dots, s_n$ drawn from the volume sampling distribution [8, Def. 1] by Markov chain Monte Carlo with optimal weights (8).

- RPCHOLESKY. Nodes $s_1, \dots, s_n$ sampled by RPCHOLESKY using algorithm 4 with the optimal weights (8).

- *Positively weighted kernel quadrature (PWKQ).* Nodes and weights computed by the Nyström+empirical+opt method as described on [22, p. 9].

See appendix D for more details about our numerical experiments. Experiments were run on a MacBook Pro with a 2.4 GHz 8-Core Intel Core i9 CPU and 64 GB 2667 MHz DDR4 RAM. Our code is available at `https://github.com/eepperly/RPCholesky-Kernel-Quadrature`.

Errors for the different methods are shown in fig. 2 (left panels). We see that RPCHOLESKY consistently performs among the best methods at every value of $n$, numerically achieving the rate

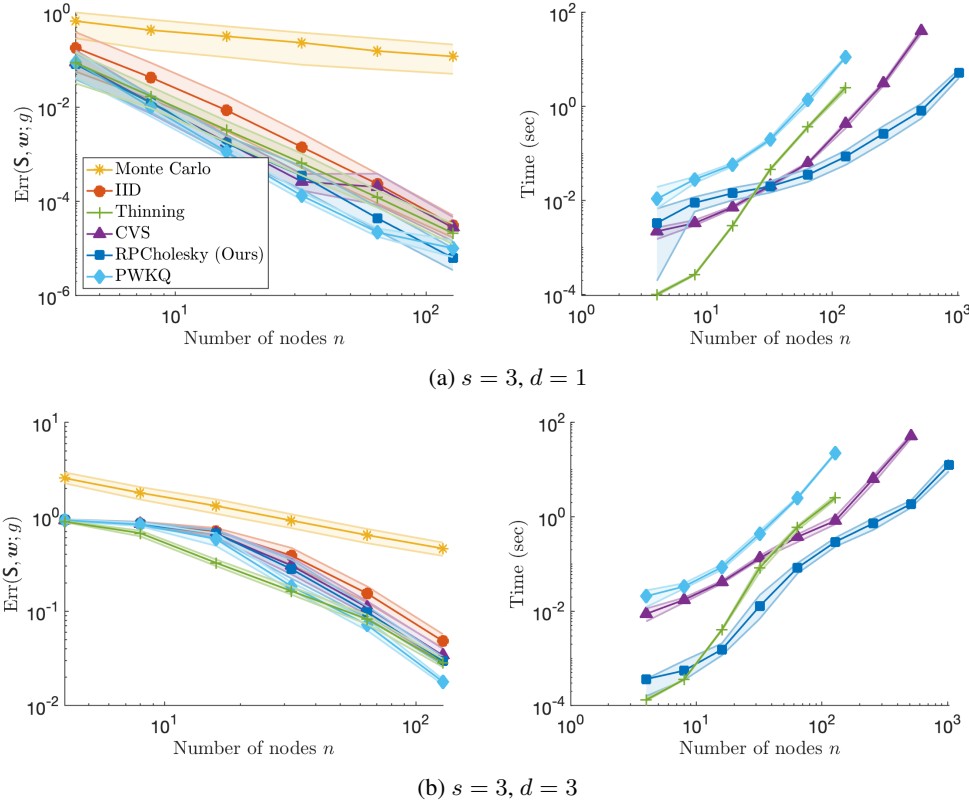

(a) $s = 3, d = 1$

(b) $s = 3, d = 3$

Figure 2: **Benchmark example: Sobolev space.** Mean quadrature error $\mathrm{Err}(\mathsf{S}, \boldsymbol{w}; g)$ (*left*) and sampling time (*right*) for different methods (100 trials) for $s = 1$, $d = 3$ (*top*) and $s = d = 3$ (*bottom*). Shaded regions show 10%/90% quantiles.

of convergence of the fastest method for each problem. Particularly striking is the result that RPCHOLESKY sampling's performance is either very close to or better than that of continuous volume sampling, despite the latter's slightly stronger theoretical properties.

Figure 2 (right panels) show the sampling times for each of the nonuniform sampling methods. RPC-HOLESKY sampling is the fastest by far. To sample 128 nodes for $s = 3$ and $d = 1$, RPCHOLESKY was **17× faster than continuous volume sampling, 52× faster than thinning, and 170× faster than PWKQ.**

To summarize our numerical evaluation (comprising fig. 2 and fig. 1 from the introduction), we find that RPCHOLESKY is among the most accurate kernel quadrature methods tested. To us, the strongest benefits of RPCHOLESKY (supported by these experiments) are the method's speed and flexibility. These virtues make it possible to apply spectrally accurate kernel quadrature in scenarios where it would have been computationally intractable before.

## 6 Application: Analyzing large chemistry datasets

While our focus thusfar has been on infinite domains $\mathsf{X}$, the kernel quadrature formalism can also be applied to a finite set $\mathsf{X}$ of data points. Let $\mu = \mathrm{UNIF}\,\mathsf{X}$ be the uniform distribution and set $g \equiv 1$. Here, the task is to exhibit $n$ nodes $\mathsf{S} = \{s_1, \ldots, s_n\} \subseteq \mathsf{X}$ and weights $\boldsymbol{w} \in \mathbb{R}^n$ such that the average of every "smooth" function $f : \mathsf{X} \to \mathbb{R}$ over the whole dataset is well-approximated by a sum:

$$\mathrm{mean}(f) := \frac{1}{|\mathsf{X}|} \sum_{x \in \mathsf{X}} f(x) \approx \sum_{i=1}^{n} w_i f(s_i). \tag{14}$$

Here is an application to chemistry. Let $\mathsf{X}$ denote a large set of compounds of interest, and let $f : \mathsf{X} \to \mathbb{R}$ denote a target chemical property such as specific heat capacity. We are interested

in computing $\mathrm{mean}(f)$. Unfortunately, evaluating $f$ on each $x \in \mathsf{X}$ requires an expensive density functional theory computation, so it can be prohibitively expensive to evaluate $f$ on every $x \in \mathsf{X}$. Fortunately, we can obtain fast approximations to $\mathrm{mean}(f)$ using (14), which only require evaluating $f$ on a much smaller set of size $|\mathsf{S}| \ll |\mathsf{X}|$.

Our experimental setup is as follows. For $\mathsf{X}$, we use $2 \times 10^4$ randomly selected points from the QM9 dataset [33, 37, 40]. We represent each compound as a vector in $\mathbb{R}^{1500}$ using the many-body tensor representation [25] computed with the DScribe package [24]. Choose $k$ to be a Gaussian kernel with bandwidth chosen by median heuristic [20] on a further subsample of $10^3$ random points. We omit continuous volume sampling, thinning, and positively weighted kernel quadrature because of their computational cost. In their place, we add the greedy Nyström method [19] with optimal weights (8).

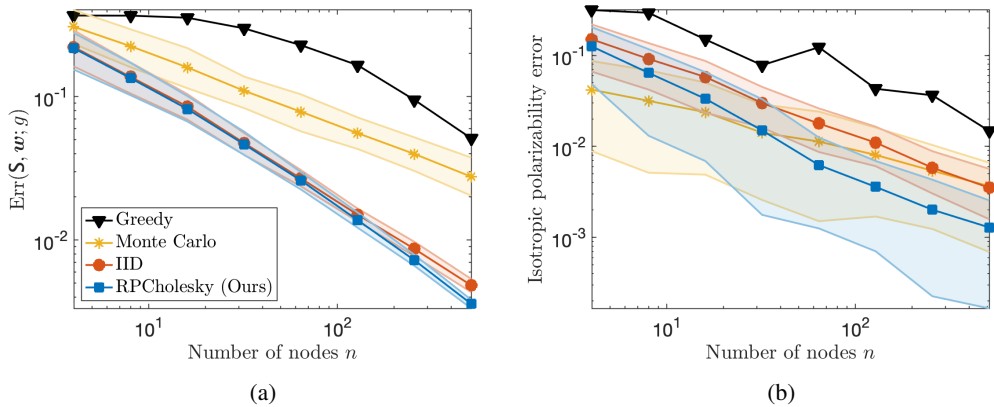

(a)                                                       (b)

Figure 3: **Application: chemistry.** Worst-case quadrature error (*left*) and mean relative error for estimation of the average value of the isotropic polarizability function $f(x)$ (*right*) for different methods (100 trials). Shaded regions show 10%/90% quantiles.

Results are shown in fig. 3. Figure 3a shows the worst-case quadrature error (9). For this metric, IID and randomly pivoted Cholesky are the definitive winners, with randomly pivoted Cholesky being slightly better. Figure 3b shows the mean relative error for the kernel quadrature estimates of the mean *isotropic polarizability* of the compounds in $\mathsf{X}$. For sufficiently large $n$, randomly pivoted Cholesky has the smallest error, beating IID and Monte Carlo by a factor of three at $n = 512$.

# 7 Conclusions, Limitations, and Possibilites for Future Work

In this article, we developed continuous RPCHOLESKY sampling for kernel quadrature. Theorem 1 demonstrates RPCHOLESKY kernel quadrature achieves near-optimal error rates. Numerical results (fig. 2) hint that its practical performance might be even better than that suggested by our theoretical analysis and fully comparable with kernel quadrature based on the much more computationally expensive continuous volume sampling distribution. RPCHOLESKY supports performant rejection sampling algorithms (algorithms 2 and 4), which facilitate implementation for general spaces $\mathsf{X}$, measures $\mu$, and kernels $k$ with ease.

We highlight three limitations of RPCHOLESKY kernel quadrature that would be worth addressing in future work. First, given the comparable performance of RPCHOLESKY and continuous volume sampling in practice, it would be desirable to prove stronger error bounds for RPCHOLESKY sampling or find counterexamples which demonstrate a separation between the methods. Second, it would be worth developing improved sampling algorithms for RPCHOLESKY which avoid the need for global optimization steps. Third, all known spectrally accurate kernel quadrature methods require integrals of the form $\int_{\mathsf{X}} k(x,y)g(y)\,\mathrm{d}\mu(y)$, which may not be available. RPCHOLESKY kernel quadrature requires them for the computation of the weights (8). Developing spectrally accurate kernel quadrature schemes that avoid such integrals remains a major open problem for the field.

## Acknowledgements

We thank Lester Mackey, Eliza O'Reilly, Joel Tropp, and Robert Webber for helpful discussions and feedback.

*Disclaimer.* This report was prepared as an account of work sponsored by an agency of the United States Government. Neither the United States Government nor any agency thereof, nor any of their employees, makes any warranty, express or implied, or assumes any legal liability or responsibility for the accuracy, completeness, or usefulness of any information, apparatus, product, or process disclosed, or represents that its use would not infringe privately owned rights. Reference herein to any specific commercial product, process, or service by trade name, trademark, manufacturer, or otherwise does not necessarily constitute or imply its endorsement, recommendation, or favoring by the United States Government or any agency thereof. The views and opinions of authors expressed herein do not necessarily state or reflect those of the United States Government or any agency thereof.

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

# A  Proof of theorem 1

We start off by proving proposition 3:

*Proof of proposition 3.* Equation (12) follows from a direct calculation:

$$\sum_{i=1}^{n} w_i k(\cdot, s_i) = k(\cdot, \mathsf{S}) \boldsymbol{w} = \int_{\mathsf{X}} k(\cdot, \mathsf{S}) k(\mathsf{S}, \mathsf{S})^{-1} k(\mathsf{S}, x) g(x) \, \mathrm{d}\mu(x) = T_{\mathsf{S}} g.$$

To establish the second claim, use the fact [3, §2.1] that $T^{-1/2} : \mathcal{H} \to \mathsf{L}^2(\mu)$ is an isometry and invoke the definition (9) of $\mathrm{Err}(\mathsf{S}; g)$:

$$\begin{aligned}
\mathrm{Err}^2(\mathsf{S}; g) = \|(T - T_{\mathsf{S}})g\|_{\mathcal{H}}^2 &= \left\| T^{-1/2}(T - T_{\mathsf{S}})g \right\|_{\mathsf{L}^2(\mu)}^2 \\
&= \langle g, (T - T_{\mathsf{S}})T^{-1}(T - T_{\mathsf{S}})g \rangle_{\mathsf{L}^2(\mu)} \le \langle g, (T - T_{\mathsf{S}})g \rangle_{\mathsf{L}^2(\mu)}.
\end{aligned}$$

The last inequality follows because $T - T_{\mathsf{S}}$ and $T_{\mathsf{S}} = T - (T - T_{\mathsf{S}})$ are both positive semidefinite. $\square$

For the rest of our proof of theorem 1, we employ the following notation. Let $\preceq$ denote the Loewner order for bounded, linear operators on $\mathsf{L}^2(\mu)$. That is, $A \preceq B$ if $B - A$ is positive semidefinite (psd). Let $k_i$ denote the residual

$$k_i := k - k_{\{s_1, \dots, s_i\}}$$

for the approximate kernel associated with the first $i$ nodes. (See proposition 8 below.) The associated integral operator is

$$T_i f := \int_{\mathsf{X}} k_i(\cdot, x) f(x) \, \mathrm{d}\mu(x).$$

To analyze RPCHOLESKY, we begin by analyzing a single step of the procedure. The expected value of the residual kernel after one step, $k_1$, is

$$\mathbb{E} \, k_1(x, y) = k(x, y) - \frac{\int_{\mathsf{X}} k(x, s) k(s, y) \, \mathrm{d}\mu(s)}{\int_{\mathsf{X}} k(z, z) \, \mathrm{d}\mu(z)}.$$

Consequently, the expected value of the integral operator $T_1$ is

$$\Phi(T) := \mathbb{E}[T_1] = T - \frac{T^2}{\mathrm{Tr}\, T}. \tag{15}$$

The map $\Phi$ enjoys several useful properties [10, Lem. 3.2].

**Proposition 5.** *The map $\Phi$ is positive, monotone, and concave with respect to the order $\preceq$ on the set of trace-class psd operators on $\mathsf{L}^2(\mu)$. That is, for trace-class psd operators $A, B$,*

$$0 \preceq A \preceq B \implies 0 \preceq \Phi(A) \preceq \Phi(B) \tag{16}$$

*and, for $\theta \in [0, 1]$,*

$$\theta \Phi(A) + (1 - \theta)\Phi(B) \preceq \Phi(\theta A + (1 - \theta)B). \tag{17}$$

*Proof.* We reproduce the proof from [10, Lem. 3.2] for completeness.

***Positivity.*** Assume $A \succeq 0$. Observe that $\mathrm{Id} \succeq A / \mathrm{Tr}\, A$, where $\mathrm{Id}$ denotes the identity operator. Thus

$$\Phi(A) = A \left( \mathrm{Id} - \frac{A}{\mathrm{Tr}\, A} \right) \succeq 0,$$

This establishes that $\Phi$ is positive: $A \succeq 0$ implies $\Phi(A) \succeq 0$.

***Concavity.*** For $\theta \in [0, 1]$, denote $\bar{\theta} := 1 - \theta$. Then

$$\Phi(\theta A + \bar{\theta} B) - \theta \Phi(A) - \bar{\theta} \Phi(B) = \frac{\theta \bar{\theta}}{\theta \, \mathrm{Tr}\, A + \bar{\theta} \, \mathrm{Tr}\, B} \left( \sqrt{\frac{\mathrm{Tr}\, B}{\mathrm{Tr}\, A}} A - \sqrt{\frac{\mathrm{Tr}\, A}{\mathrm{Tr}\, B}} B \right) \succeq 0.$$

This establishes the concavity claim (17).

***Monotonicity.*** Suppose $0 \preceq A \preceq B$. Observe the map $\Phi$ is positive homogeneous: For $\theta > 0$, $\Phi(\theta A) = \theta \Phi(A)$. Thus,

$$\Phi(B) = \Phi(A + (B - A)) = 2\Phi\left(\frac{A + (B - A)}{2}\right).$$

Invoke concavity (17) to obtain

$$\Phi(B) = 2\Phi\left(\frac{A + (B - A)}{2}\right) \succeq \Phi(A) + \Phi(B - A) \succeq \Phi(A).$$

In the last inequality, we use the positivity $\Phi(B - A) \succeq 0$. This completes the proof of (16). $\qquad\square$

We can use the map $\Phi$ to bound the residual integral operator $T_i$:

**Proposition 6.** *For $i \geq 0$, we have the bound*

$$\mathbb{E}[T_i] \preceq \Phi^i(T), \tag{18}$$

*where $\Phi^i$ denotes the $i$-fold composition of $\Phi$.*

*Proof.* The $i$th step of RPCHOLESKY applies one step of RPCHOLESKY to $k_{i-1}$. Thus,

$$\mathbb{E}[T_i] = \mathbb{E}[\mathbb{E}[T_i \mid s_1, \dots, s_{i-1}]] = \mathbb{E}[\Phi(T_{i-1})].$$

Now, we can apply concavity (17) to obtain

$$\mathbb{E}[T_i] = \mathbb{E}[\Phi(T_{i-1})] \preceq \Phi(\mathbb{E}[T_{i-1}]).$$

Applying this result twice and using monotonicity (16) yields:

$$\mathbb{E}[T_i] \preceq \Phi(\mathbb{E}[T_{i-1}]) \text{ and } \mathbb{E}[T_{i-1}] \preceq \Phi(\mathbb{E}[T_{i-2}]) \implies \mathbb{E}[T_i] \preceq \Phi(\Phi(\mathbb{E}[T_{i-2}])).$$

Iterating this argument, we obtain the desired conclusion (18). $\qquad\square$

We now have the tools to prove theorem 1. We actually prove the following slight refinement:

**Theorem 7.** *Let $\mathsf{S} = \{s_1, \dots, s_n\}$ be generated by the RPCHOLESKY sampling algorithm. For any function $g \in \mathsf{L}^2(\mu)$, nonnegative integer $r \geq 0$, and real numbers $a, \varepsilon > 0$, we have*

$$\mathbb{E}\,\mathrm{Err}^2(\mathsf{S}; g) \leq \|g\|^2_{\mathsf{L}^2(\mu)} \cdot \left(a + \varepsilon \sum_{i=r+1}^{\infty} \lambda_i\right)$$

*provided*

$$n \geq r \log\left(\frac{\lambda_1}{a}\right) + \frac{1}{\varepsilon}. \tag{19}$$

Theorem 1 immediately follows from theorem 7 by taking $\varepsilon = \delta/2$ and $a = (\delta/2)\sum_{i=r+1}^{\infty} \lambda_i$. We now prove theorem 7.

*Proof of theorem 7.* In the standing notation, $T - T_\mathsf{S}$ corresponds to $T_n$. Thus, by proposition 6, we have

$$\mathbb{E}\langle g, (T - T_\mathsf{S})g\rangle_{\mathsf{L}^2(\mu)} = \langle g, \mathbb{E}[T_n]g\rangle_{\mathsf{L}^2(\mu)} \leq \langle g, \Phi^n(T)g\rangle_{\mathsf{L}^2(\mu)} \leq \|g\|^2_{\mathsf{L}^2(\mu)} \cdot \lambda_{\max}(\Phi^n(T)).$$

Thus, it is sufficient to show that

$$\lambda_{\max}(\Phi^n(T)) \leq a + \varepsilon \sum_{i=r+1}^{\infty} \lambda_i \tag{20}$$

holds under condition (19).

Using the definition (15) of $\Phi$, we see that the eigenvalues $\lambda_1^{(i)} \geq \lambda_2^{(i)} \geq \cdots$ of $\Phi^i(T)$ are given by the following recurrence

$$\lambda_j^{(i)} = \lambda_j^{(i-1)} - \frac{\left(\lambda_j^{(i-1)}\right)^2}{\sum_{\ell=1}^{\infty} \lambda_\ell^{(i-1)}} \quad \text{for } i, j = 1, 2, \dots \tag{21}$$

with initial condition $\lambda_j^{(0)} = \lambda_j$ for all $j$. From this formula, we immediately conclude that each $\lambda_j^{(i)}$ is nonnegative and nonincreasing in $i$. In addition, the ordinal ranking $\lambda_1^{(i)} \geq \lambda_2^{(i)} \geq \cdots$ is preserved for each $i$. Indeed, if $\lambda_{j+1}^{(i-1)} - \lambda_j^{(i-1)} \geq 0$, then

$$
\begin{aligned}
\lambda_{j+1}^{(i)} - \lambda_j^{(i)} &= \lambda_{j+1}^{(i-1)} - \lambda_j^{(i-1)} - \frac{\left(\lambda_{j+1}^{(i-1)}\right)^2 - \left(\lambda_j^{(i-1)}\right)^2}{\sum_{\ell=1}^{\infty} \lambda_\ell^{(i-1)}} \\
&= \left(\lambda_{j+1}^{(i-1)} - \lambda_j^{(i-1)}\right)\left(1 - \frac{\lambda_{j+1}^{(i-1)} + \lambda_j^{(i-1)}}{\sum_{\ell=1}^{\infty} \lambda_\ell^{(i-1)}}\right) \geq 0,
\end{aligned}
$$

and the claim follows by induction on $i$. To bound $\lambda_{\max}(\Phi^i(T)) = \lambda_1^{(i)}$, we bound the sum of eigenvalues appearing in the recurrence (21):

$$
\sum_{\ell=1}^{\infty} \lambda_\ell^{(i-1)} = \lambda_1^{(i-1)} + \cdots + \lambda_r^{(i-1)} + \sum_{\ell=r+1}^{\infty} \lambda_\ell^{(i-1)} \leq r\lambda_1^{(i-1)} + \sum_{\ell=r+1}^{\infty} \lambda_\ell. \tag{22}
$$

To bound the first $r$ terms, we used the ordering of the eigenvalues $\lambda_1^{(i-1)} \geq \cdots \geq \lambda_r^{(i-1)}$. To bound the tail, we used the fact that for each $\ell \in \{r+1, r+2, \ldots\}$, the eigenvalue $\lambda_\ell^{(i-1)}$ is nonincreasing in $i$ and thus bounded by $\lambda_\ell^{(0)} = \lambda_\ell$. Substituting (22) in (21), we obtain

$$
\lambda_1^{(i)} \leq \lambda_1^{(i-1)} - \frac{\left(\lambda_1^{(i-1)}\right)^2}{r\lambda_1^{(i-1)} + \sum_{\ell=r+1}^{\infty} \lambda_\ell} \quad \text{for } i = 1, 2, \ldots. \tag{23}
$$

To bound the solution to the recurrence (23), we pass to continuous time. Specifically, for any $t \in \mathbb{Z}_+$, $\lambda_1^{(t)}$ is bounded by the process $x(t)$ solving the initial value problem

$$
\frac{\mathrm{d}}{\mathrm{d}t} x(t) = -\frac{x(t)^2}{rx(t) + \sum_{\ell=r+1}^{\infty} \lambda_\ell}, \quad x(0) = \lambda_1.
$$

The bound $\lambda_1^{(t)} \leq x(t)$ holds for all $t \in \mathbb{Z}_+$ because $x \mapsto -x^2/(rx + \sum_{\ell=r+1}^{\infty} \lambda_\ell)$ is decreasing for $x \in (0, \infty)$. We are interested in the time $t = t_\star$ at which $x(t_\star) = a + \varepsilon \sum_{\ell=r+1}^{\infty} \lambda_\ell$, which we obtain by separation of variables

$$
\begin{aligned}
t_\star &= \int_{\lambda_1}^{a+\varepsilon \sum_{\ell=r+1}^{\infty} \lambda_\ell} -\frac{rx + \sum_{\ell=r+1}^{\infty} \lambda_\ell}{x^2} \, \mathrm{d}x = \left[-r \log x + \frac{\sum_{\ell=r+1}^{\infty} \lambda_\ell}{x}\right]_{x=\lambda_1}^{x=a+\varepsilon \sum_{\ell=r+1}^{\infty} \lambda_\ell} \\
&= r \log \frac{\lambda_1}{a + \varepsilon \sum_{\ell=r+1}^{\infty} \lambda_\ell} + \frac{\sum_{\ell=r+1}^{\infty} \lambda_\ell}{a + \varepsilon \sum_{\ell=r+1}^{\infty} \lambda_\ell} - \frac{\sum_{\ell=r+1}^{\infty} \lambda_\ell}{\lambda_1} \leq r \log \frac{\lambda_1}{a} + \frac{1}{\varepsilon}.
\end{aligned}
$$

Since $x$ is decreasing, the following holds for $n$ satisfying (10):

$$
\lambda_{\max}(\Phi^n(T)) = \lambda_1^{(n)} \leq x(n) \leq x(t_\star) = a + \varepsilon \sum_{\ell=r+1}^{\infty} \lambda_\ell.
$$

This shows (20), completing the proof. $\qquad\square$

## B  Proof of theorem 4

To see why algorithm 2 produces samples from the RPCHOLESKY sampling distribution, observe that at each effective iteration $i$ the matrix $\boldsymbol{L}(1 : i - 1, 1 : i - 1)$ is the Cholesky factor of $k(\mathsf{S}, \mathsf{S})$. Therefore, $d$ is equal to the residual kernel $k_{\mathsf{S}}^{\mathrm{res}} := k - k_{\mathsf{S}}$ evaluated at $s_i$

$$
d = k_{\mathsf{S}}^{\mathrm{res}}(s_i, s_i).
$$

At each effective iteration $i$, the candidate point $s_i$ is sampled from $k(x,x)\,\mathrm{d}\mu(x)$ and accepted with probability $k_{\mathsf{S}}^{\mathrm{res}}(s_i, s_i)/k(s_i, s_i)$. Therefore, conditional on acceptance and previous steps of the algorithm, $s_i$ is sampled from the probability measure

$$\frac{1}{\mathbb{P}\{\text{Acceptance} \mid \mathsf{S}\}} \cdot k(x,x)\,\mathrm{d}\mu(x) \cdot \frac{k_{\mathsf{S}}^{\mathrm{res}}(x,x)}{k(x,x)} = \frac{k_{\mathsf{S}}^{\mathrm{res}}(x,x)\,\mathrm{d}\mu(x)}{\mathbb{P}\{\text{Acceptance} \mid \mathsf{S}\}}.$$

Since this is a probability measure which must integrate to one, the probability of acceptance is

$$\mathbb{P}\{\text{Acceptance} \mid \mathsf{S}\} = \int_{\mathsf{X}} k_{\mathsf{S}}^{\mathrm{res}}(x,x)\,\mathrm{d}\mu(x).$$

Therefore, conditional on acceptance and previous steps of the algorithm, $s_i$ is sampled from the probability measure $k_{\mathsf{S}}^{\mathrm{res}}(x,x)\,\mathrm{d}\mu(x)/\int_{\mathsf{X}} k_{\mathsf{S}}^{\mathrm{res}}(x,x)\,\mathrm{d}\mu(x)$. This corresponds exactly with the RPCHOLESKY sampling procedure described in algorithm 1 (see proposition 8 below).

Now, we bound the runtime. For each iteration $i$, the expected acceptance probability is bounded from below by the smallest possible trace error achievable by a rank $(i-1)$ approximation

$$\mathbb{P}\{\text{Acceptance}\} = \mathbb{E}[\mathbb{P}\{\text{Acceptance} \mid \mathsf{S}\}] = \mathbb{E}\left[\int_{\mathsf{X}} k_{\mathsf{S}}^{\mathrm{res}}(x,x)\,\mathrm{d}\mu(x)\right] \geq \eta_{i-1}.$$

Therefore, the expected number of rejections at step $i$ is bounded by the expectation of a geometric random variable with parameter $\eta_{i-1}$, i.e., by $\eta_{i-1}^{-1}$. Since each loop iteration takes $\mathcal{O}(i^2)$ operations (attributable to line 4) and at most $\eta_{i-1}^{-1}$ expected iterations are required to accept at step $i$, the total expected runtime is at most $\mathcal{O}(\sum_{i=1}^{n} i^2/\eta_{i-1}) \leq \mathcal{O}(n^3/\eta_{n-1})$. $\qquad\square$

## C    Derivation of (7)

For completeness, we provide a derivation of (7). This result is standard; see, e.g., [3, Eq. (7)]. Recall our definition (6) for $\mathrm{Err}(\mathsf{S}, \boldsymbol{w}; g)$:

$$\mathrm{Err}(\mathsf{S}, \boldsymbol{w}; g) = \max_{\|f\|_{\mathcal{H}} \leq 1} \left| \int_{\mathsf{X}} f(x)g(x)\,\mathrm{d}\mu(x) - \sum_{i=1}^{n} w_i f(s_i) \right|.$$

Use the reproducing property of $k$ and linearity to write

$$\mathrm{Err}(\mathsf{S}, \boldsymbol{w}; g) = \max_{\|f\|_{\mathcal{H}} \leq 1} \left| \int_{\mathsf{X}} \langle f, k(\cdot, x)\rangle_{\mathcal{H}} g(x)\,\mathrm{d}\mu(x) - \sum_{i=1}^{n} w_i \langle f, k(\cdot, s_i)\rangle_{\mathcal{H}} \right|.$$

Now apply linearity of $\langle \cdot, \cdot\rangle_{\mathcal{H}}$ to obtain

$$\mathrm{Err}(\mathsf{S}, \boldsymbol{w}; g) = \max_{\|f\|_{\mathcal{H}} \leq 1} \left| \left\langle f, \int_{\mathsf{X}} k(\cdot, x)g(x)\,\mathrm{d}\mu(x) - \sum_{i=1}^{n} w_i k(\cdot, s_i)\right\rangle_{\mathcal{H}} \right|.$$

Use the dual characterization of the RKHS norm $\|\cdot\|_{\mathcal{H}}$ and the definition (5) of $T$ to conclude

$$\mathrm{Err}(\mathsf{S}, \boldsymbol{w}; g) = \left\| Tg - \sum_{i=1}^{n} w_i k(\cdot, s_i) \right\|_{\mathcal{H}}.$$

## D    Details for numerical experiments

In this section, we provide additional details about our numerical experiments.

**Continuous volume sampling.**    We are unaware of a provably correct $n$-DPP/continuous volume distribution sampler that is sufficiently efficient to facilitate numerical testing and comparisons. The best-known sampler [34] for the continuous setting is inexact and requires $\mathcal{O}(n^5 \log n)$ MCMC steps, which would be prohibitive for us. To produce samples from the continuous volume sampling distribution for this paper, we use a continuous analog of the MCMC algorithm [2]. Given a set $\mathsf{S} = \{s_1, \dots, s_n\}$ and assuming $\mu$ is a probability measure, one MCMC step proceeds as follows:

1. Sample $s_{\text{new}} \sim \mu$ and $s_{\text{old}} \sim \text{UNIF}\,\mathsf{S}$.
2. Define $\mathsf{S}' = \mathsf{S} \cup \{s_{\text{new}}\} \setminus \{s_{\text{old}}\}$.
3. With probability
$$\frac{1}{2} \min\left\{ 1, \frac{\det k(\mathsf{S}', \mathsf{S}')}{\det k(\mathsf{S}, \mathsf{S})} \right\},$$
accept and set $\mathsf{S} \leftarrow \mathsf{S}'$. Otherwise, leave $\mathsf{S}$ unchanged.

In our experiments, we initialize with $s_1, \ldots, s_n$ drawn iid from $\mu$ and run for $10n$ MCMC steps.

Running for just $10n$ MCMC steps is aggressive, even in the finite setting. Indeed, the theoretical analysis of [2, Thm. 2, Eq. (1.2)] for $\mu = \text{UNIF}\{1, 2, \ldots, |\mathsf{X}|\}$, proves the sampler mixes to stationarity in at most $\tilde{\mathcal{O}}(n^2 |\mathsf{X}|)$ MCMC steps when properly initialized. Even with just $10n$ MCMC steps, continuous volume sampling is dramatically more expensive than RPCHOLESKY in our experiments for $n \geq 64$.

Finally, we notice that the performance of continuous volume sampling degrades to that of iid sampling for sufficiently large $n$. We believe that the cause for this are numerical issues in evaluating the determinants of the kernel matrices $k(\mathsf{S}, \mathsf{S})$. This suggests that, in addition to being faster, RPCHOLESKY may be more numerically robust than continuous volume sampling.

**Thinning and positively weighted kernel quadrature.** Our kernel quadrature experiments with thinning and PWKQ is based on code from the authors of [22] available online at `https://github.com/satoshi-hayakawa/kernel-quadrature`, which uses the `goodpoints` package (`https://github.com/microsoft/goodpoints`) by the authors of [17, 18, 38] to perform thinning. We use $\mathfrak{g} = 4$, $\delta = 0.5$, and four bins for the COMPRESS++ algorithm.

**Miscellaneous.** Our code is available online at

`https://github.com/eepperly/RPCholesky-Kernel-Quadrature.`

To evaluate exact integrals
$$Tg(x) = \int_{\mathsf{X}} k(x, y)g(y)\, \mathrm{d}\mu(y)$$
for fig. 1, we use the 2D functionality in ChebFun [15]. To compute the optimal weights (8), we add a small multiple of the identity to regularize the system:
$$\boldsymbol{w}_{\star,\,\text{reg}} = (k(\mathsf{S}, \mathsf{S}) + 10\varepsilon_{\text{mach}} \operatorname{trace}(k(\mathsf{S}, \mathsf{S})) \cdot \mathbf{I})^{-1} Tg(\mathsf{S}).$$
Here, $\varepsilon_{\text{mach}} = 2^{-52}$ is the double precision machine epsilon.

## E  Randomly pivoted Cholesky and Nyström approximation

In this section, we describe the connection between the RPCHOLESKY algorithm and Nyström approximation. We hope that this explanation provides context that will be helpful in understanding the RPCHOLESKY algorithm and the proofs of theorems 1 and 4.

To make the following discussion more clear, we begin by rewriting the basic RPCHOLESKY algorithm (see algorithm 1) in algorithm 5 to avoid overwriting the kernel $k$ at each step:

From this rewriting, we see that in addition to selecting quadrature nodes $s_1, \ldots, s_n$, the RPC-HOLESKY algorithm builds an approximation $\hat{k}_n$ to the kernel $k$. Indeed, one can verify that the approximation $\hat{k}_n$ produced by this algorithm is precisely the Nyström approximation (definition 2).

**Proposition 8.** *The output $\hat{k}_n$ of algorithm 5 is equal to the Nyström approximation $k_{\{s_1,\ldots,s_n\}}$.*

*Proof.* Proceed by induction on $n$. The base case $n = 0$ is immediate. Fix $n$ and let $\mathsf{S} := \{s_1, \ldots, s_n\}$, $s' := s_{n+1}$, and $\mathsf{S}' := \{s_1, \ldots, s_n, s'\}$. Write the Nyström approximation $k_{\mathsf{S}'}$ as

$$k_{\mathsf{S}'} = \begin{bmatrix} k(\cdot, \mathsf{S}) & k(\cdot, s') \end{bmatrix} \begin{bmatrix} k(\mathsf{S}, \mathsf{S}) & k(\mathsf{S}, s') \\ k(s', \mathsf{S}) & k(s', s') \end{bmatrix}^{-1} \begin{bmatrix} k(\mathsf{S}, \cdot) \\ k(s', \cdot) \end{bmatrix}$$

**Algorithm 5** RPCHOLESKY: unoptimized implementation without overwriting

---

**Input:** Kernel $k : \mathsf{X} \times \mathsf{X} \to \mathbb{R}$ and number of quadrature points $n$
**Output:** Quadrature points $s_1, \ldots, s_n$ and Nyström approximation $\hat{k}_n = k_{\{s_1,\ldots,s_n\}}$ to $k$
1: Initialize $k_0 \leftarrow k$, $\hat{k}_0 \leftarrow 0$
2: **for** $i = 1, 2, \ldots, n$ **do**
3:     Sample $s_i$ from the probability measure $k(x,x)\,\mathrm{d}\mu(x)/\int_\mathsf{X} k(x,x)\,\mathrm{d}\mu(x)$
4:     Update Nyström approximation $\hat{k}_i \leftarrow \hat{k}_{i-1} + k_{i-1}(\cdot, s_i)k_{i-1}(s_i, s_i)^{-1}k_{i-1}(s_i, \cdot)$
5:     Update kernel $k_i \leftarrow k_{i-1} - k_{i-1}(\cdot, s_i)k_{i-1}(s_i, s_i)^{-1}k_{i-1}(s_i, \cdot)$
6: **end for**

---

Compute a block $\boldsymbol{L D L}^\top$ factorization of the matrix in the middle of this expression:

$$
\begin{bmatrix} k(\mathsf{S},\mathsf{S}) & k(\mathsf{S},s') \\ k(s',\mathsf{S}) & k(s',s') \end{bmatrix} = \begin{bmatrix} \mathbf{I} & \mathbf{0} \\ k(s',\mathsf{S})k(\mathsf{S},\mathsf{S})^{-1} & 1 \end{bmatrix} \begin{bmatrix} k(\mathsf{S},\mathsf{S}) & \mathbf{0} \\ \mathbf{0}^\top & k_n(s',s') \end{bmatrix} \begin{bmatrix} \mathbf{I} & k(\mathsf{S},\mathsf{S})^{-1}k(\mathsf{S},s') \\ \mathbf{0}^\top & 1 \end{bmatrix}.
$$

Here, we've used the induction hypothesis

$$
k_n(s',s') = k(s',s') - k_\mathsf{S}(s',s') = k(s',s') - k(s',\mathsf{S})k(\mathsf{S},\mathsf{S})^{-1}k(\mathsf{S},s'). \tag{24}
$$

Thus,

$$
\begin{aligned}
k_{\mathsf{S}'} &= [k(\cdot,\mathsf{S}) \quad k(\cdot,s')] \begin{bmatrix} k(\mathsf{S},\mathsf{S}) & k(\mathsf{S},s') \\ k(s',\mathsf{S}) & k(s',s') \end{bmatrix}^{-1} \begin{bmatrix} k(\mathsf{S},\cdot) \\ k(s',\cdot) \end{bmatrix} \\
&= [k(\cdot,\mathsf{S}) \quad k(\cdot,s')] \begin{bmatrix} \mathbf{I} & -k(\mathsf{S},\mathsf{S})^{-1}k(\mathsf{S},s') \\ \mathbf{0}^\top & 1 \end{bmatrix} \begin{bmatrix} k(\mathsf{S},\mathsf{S})^{-1} & \mathbf{0} \\ \mathbf{0}^\top & k_n(s',s')^{-1} \end{bmatrix} \\
&\qquad \cdot \begin{bmatrix} \mathbf{I} & \mathbf{0} \\ -k(s',\mathsf{S})k(\mathsf{S},\mathsf{S})^{-1} & 1 \end{bmatrix} \begin{bmatrix} k(\mathsf{S},\cdot) \\ k(s',\cdot) \end{bmatrix} \\
&= [k(\cdot,\mathsf{S}) \quad k_n(\cdot,s')] \begin{bmatrix} k(\mathsf{S},\mathsf{S})^{-1} & \mathbf{0} \\ \mathbf{0}^\top & k_n(s',s')^{-1} \end{bmatrix} \begin{bmatrix} k(\mathsf{S},\cdot) \\ k_n(\cdot,s') \end{bmatrix} \tag{25} \\
&= \hat{k}_n + k_n(\cdot,s')k_n(s',s')^{-1}k_n(s',\cdot) = \hat{k}_{n+1}.
\end{aligned}
$$

For (25), we used the induction hypothesis (24) again. Having established that $k_{\mathsf{S}'} = k_{n+1}$, the proposition is proven. $\qquad\square$