# OpenReview forum: "Kernel Quadrature with Randomly Pivoted Cholesky"
_NeurIPS.cc/2023/Conference — NeurIPS 2023 spotlight_

### Official Review · Reviewer_PPmk · 2023-06-29

**Soundness:** 3 good
**Presentation:** 3 good
**Contribution:** 3 good
**Rating:** 6
**Confidence:** 3

**Summary:**

The kernel quadrature problem computes quadrature weights and nodes that minimize the maximum quadrature error over all functions in RKHS with the norm bounded by 1. This paper proposes new quadrature rules for functions in RKHS using nodes drawn by randomly pivoted Cholesky algorithm. The randomly pivoted Cholesky used in the paper is an extension of the finite variant introduced in [10]. The authors provide the unoptimized implementation followed by the variant with efficient rejection sampling.

**Strengths:**

1. Quadrature problem is one of the fundamental problems in computational mathematics and the paper proposes an efficient method for it.

2. The experiments offer comprehensive comparisons against other state-of-the-art methods.

3. Theorem 1 provides error bounds which achieve spectral accuracy with just logarithmic number of nodes.

4. The proposed method (both the unoptimized and the optimized variants) is easy to implement from the description.

**Weaknesses:**

1. There are no empirical results on real datasets despite earlier work such as [19] having results on real datasets.

2. Discussion on the application of the proposed method in machine learning is largely missing.

**Questions:**

1. For the right side of the Figure 2 (time vs number of nodes): what is the error level considered for all methods?

2. Could the authors comment on how the results could change for other Mercer kernels?

3. How about the effect of dimensionality on the empirical results? The reference [19] does offer more parameter settings used for evaluation.

4. Seeing that Algorithm 2 and Algorithm 3 stores Cholesky factors, could the authors comment on scalability of the method as the number of quadrature points is increased?

**Limitations:**

There are no results on real datasets and even on synthetic datasets, the results are limited to one type of kernel. Although it is clear the proposed method has wide applications in machine learning, the authors should demonstrate a usecase in which the proposed method can be applied (e.g. scaling kernel machines).

---

> ### Author Rebuttal · Authors · 2023-08-09
>
> We thank the reviewer for their careful reading and their many insightful questions and suggestions. Our responses are as follows:
>
> ## Applications
>
> The new version of the paper discusses applications more and includes a new experiment demonstrating an application to analyzing large chemistry datasets. (See the response to all reviewers for a description of our experiment and Fig. 1 in the attached PDF for the results.) We hope this example is both simple (so that it may be appreciated by a wide audience) and convincing (the QM9 dataset we use is considered a standard, but challenging, benchmark in chemical machine learning). Many of the other applications of kernel quadrature to machine learning (such as the applications we cite to reinforcement learning [27]) are sufficiently intricate that it would be difficult to briefly summarize, but we mention them and provide references.
>
> ## Questions
>
> Below, we answer each of the reviewer’s questions:
>
> > For the right side of the Figure 2 (time vs number of nodes): what is the error level considered for all methods?
>
> We are not certain exactly how to interpret the reviewer’s question. We will do our best to answer based on our interpretation of the question (and hope that we can provide a better answer, if needed, in the discussion period). For a given number of nodes, Fig. 2 (left) shows the error and Fig. 2 (right) shows the runtime. We do not specify a particular error level for Fig. 2 (right).
>
> > Could the authors comment on how the results could change for other Mercer kernels?
>
> As we demonstrate in Thm. 1, the performance of RPCholesky kernel quadrature depends on the eigenvalues of the kernel (as revealed by its Mercer decomposition (2)). For generic domains equipped with the Matérn (Fig. 1) or Sobolev (Fig. 2) kernels, the eigenvalue decay is polynomial and thus the quadrature error decays at a polynomial rate as well. For a Gaussian kernel (Fig. 1, attached PDF) the generic eigenvalue decay is exponential and thus the quadrature error decays at an exponential rate. In high dimensions, the asymptotic convergence regime may take exponentially large numbers of quadrature nodes to reach, and the kernel eigenvalues can depend sensitively on the ground space $\mathsf{X}$. Whatever the eigenvalues are, RPCholesky achieves near-optimal quadrature error and is, in our empirical tests, the fastest spectrally accurate kernel quadrature scheme.
>
> > How about the effect of dimensionality on the empirical results? The reference [19] does offer more parameter settings used for evaluation.
>
> While the reference [19] considers a few more parameter settings for the Sobolev example, they are all restricted to the same values $s,d\le 3$ that we consider in the present paper. It is difficult to compare all methods for large dimensions $d$, as larger values of $n$ are needed before the error starts to show any decrease. These larger values of $n$ are hard to reach using more expensive sampling methods like thinning and PWKQ. Indeed, we did some experiments with $d=5$ and $s=8$ using a subset of the methods; see the middle and right panels of Fig. 3 in the attached PDF. RPCholesky is noticeably—if modestly—better than CVS and IID. Given the time constraints of the NeurIPS response window, we were unable to include the more time-expensive thinning and PWKQ examples for this problem.
>
> We also draw the reviewer’s attention to our new application example (Fig. 1 in the attached PDF). This demonstrates that RPCholesky kernel quadrature extends gracefully to a 1,500-dimensional application example.
>
> > Seeing that Algorithm 2 and Algorithm 3 stores Cholesky factors, could the authors comment on scalability of the method as the number of quadrature points is increased?
>
> All optimally weighted kernel quadrature methods require the solution of a linear system of equations $k(\mathsf{S},\mathsf{S})\boldsymbol{w}_\star = Tg(\mathsf{S})$, requiring a Cholesky factorization of $k(\mathsf{S},\mathsf{S})$ at $O(n^2)$ space and $O(n^3)$ time. So, by this metric, RPCholesky’s resource requirements are no more than any optimally weighted kernel quadrature scheme. In our review of the kernel literature, existing investigations have focused on $n\lessapprox 1000$, so the requirements of storing and factoring $k(\mathsf{S},\mathsf{S})$ have not come to be a true bottleneck.
>
> Should we be interested in pushing to larger values of $n$, we suspect that RPCholesky kernel quadrature (and indeed other optimally weighted kernel quadrature schemes) could be accelerated by using fast solvers developed in the kernel algorithms community. In place of inversion or full Cholesky factorization of $k(\mathsf{S},\mathsf{S})$, natural approaches in low dimensions are hierarchical low-rank decompositions (Ambikasaran et al.) or sparsified Cholesky factorizations (Schäfer, Katzfuss, and Owhadi). For high dimensions, Nyström preconditioning (Frangella, Tropp, and Udell) could be an effective tool to compute the action of $k(\mathsf{S},\mathsf{S})^{-1}$ (which is all that is needed to implement RPCholesky kernel quadrature).
>
> Ambikasaran, S., Foreman-Mackey, D., Greengard, L., Hogg, D. W., & O’Neil, M. (2016). Fast Direct Methods for Gaussian Processes. IEEE Transactions on Pattern Analysis and Machine Intelligence, 38(2), 252–265. https://doi.org/10.1109/TPAMI.2015.2448083
>
> Frangella, Z., Tropp, J. A., & Udell, M. (2021). Randomized Nyström Preconditioning. ArXiv:2110.02820 [Cs, Math]. http://arxiv.org/abs/2110.02820
>
> Schäfer, F., Katzfuss, M., & Owhadi, H. (2021). Sparse Cholesky Factorization by Kullback–Leibler Minimization. SIAM Journal on Scientific Computing, A2019–A2046. https://doi.org/10.1137/20M1336254.

---

> > ### Comment · Reviewer_PPmk · 2023-08-14
> > **Thanks to the authors for the feedback**
> >
> > The feedback I received from the authors is very helpful in clarifying some parts of the paper. In particular, the QM dataset application (chemical engineering) is an interesting application which could strengthen the paper. I think in order to gain more tractability for general machine learning applications, the authors should pursue how to scale to larger values of $n$. I will maintain my current score as the authors addressed all of my concerns.

---

### Official Review · Reviewer_D4rs · 2023-07-03

**Soundness:** 4 excellent
**Presentation:** 4 excellent
**Contribution:** 3 good
**Rating:** 8
**Confidence:** 4

**Summary:**

This paper introduces a new method for kernel quadrature which relies on a randomly pivoted Cholesky decomposition.
The sampling of the quadrature points is done thanks to a rejection sampling algorithm. To improve performance, an optimized rejection sampling procedure is also introduced, so that the acceptance probability is updated if the number of rejections becomes too large. Numerical simulations show that this method is fast and often as accurate as other performant methods.
An error bound for the expectation of the quadrature method is also derived.

**Strengths:**

- This paper is very well written. It mentions other important methods for kernel quadrature.
- The main idea of this research work is simple and elegant.
- The theoretical results are well introduced and the proofs are clear although non-trivial.
- From the practical perspective, the numerical simulations are convincing. There is a clear gain in sampling time, in particular with respect to continuous volume sampling which is a closely related method.

**Weaknesses:**

There are very few weaknesses in this paper.
- It seems that the proof techniques are mainly inspired from [10] to which the authors refer. This is not an issue and does not limit the significance of this contribution.
- The end of the proof of proposition 3 is not very explicit. However, the authors provide the main elements to finish the proof so that the reader can complete the argument on a sheet of paper.



**Questions:**

- Concerning the simulations reported in Fig 2 (a) (left-hand side), I wonder if there is a numerical issue in the computation of the curve corresponding to continuous volume sampling (CVS). It seems to me that CVS should have at least a similar performance as RPCholesky.
However, for the last two data points, CVS suddenly has a worse performance (following the IID case). This phenomenon is not happening in Fig 2 (b) (left-hand side). Hence, I wonder if there could be an issue in the implementation of CVS such as a numerical instability. Could you comment about that ?


**Limitations:**

The authors mention possible limitations.

---

> ### Author Rebuttal · Authors · 2023-08-09
>
> We thank the reviewer for their thoughtful feedback, and we are delighted that they appreciated the paper. Below, we address the reviewer’s comments and questions:
>
> ## Proof of Proposition 3
>
> At the suggestion of yiWH, the proof of proposition 3 has been deferred to the supplementary material. We provide a more explicit proof to make this result easier to read and understand.
>
> ## CVS Numerical Performance
>
> The reviewer very astutely points out that CVS sampling degrades to the error of IID sampling in Fig. 2(a). We observed this too and found it very curious. Due to space reasons, we commented on this in Appendix C in the supplementary materials, writing:
>
> > Finally, we notice that the performance of continuous volume sampling degrades to that of iid sampling for sufficiently large $n$. We believe that the cause for this are numerical issues in evaluating the determinants of the kernel matrices $k(\mathsf{S},\mathsf{S})$. This suggests that, in addition to being faster, RPCholesky may be more numerically robust than continuous volume sampling.
>
> Numerical issues with DPP samplers, continuous and finite, inexact and exact, seem to be a widespread issue. In our experience with both our implementation and DPPy, DPP samplers frequently fail or suffer performance degradations due to numerical issues. It is possible these numerical issues could be resolved with further research attention.

---

> > ### Comment · Reviewer_D4rs · 2023-08-14
> >
> > This answers my questions. Thank you.

---

### Official Review · Reviewer_yiWH · 2023-07-19

**Soundness:** 3 good
**Presentation:** 3 good
**Contribution:** 2 fair
**Rating:** 6
**Confidence:** 3

**Summary:**

The authors provide theoretical analysis and show that RPCholesky has near optimal error rates.
Meanwhile, the authors propose to combine RPCholesky with rejection sampling in order to use it in the continuous setting. Experiments show that the proposed method is compared favorably to continuous volume sampling which is much more difficult to sample.

**Strengths:**

The paper is clearly written.

Both assumptions and theorems are well presented and discussed.

The experiments demonstrate the good performance of the proposed method.

**Weaknesses:**

For ''Err'' under line 140 and ''Err'' under line 141, it’s better to include the derivation in the appendix to make the paper self-contained.

The description in Line 226 and Line 227 is unclear. For instance, “opt” is imprecise.

The proof of the theorem in Section 3.4 can be moved to the appendix. It seems not necessary to include it in the main paper.

I think the biggest weakness of the paper is: the experiments are very limited. I would like to see more experiments as the authors claim that their method is very general.

**Questions:**

Can the authors include additional experiments and show that the proposed method can also perform well in practice, e.g. real-world experiments? Is this something difficult to implement?

**Limitations:**

The limitations are discussed in Section 6.

---

> ### Author Rebuttal · Authors · 2023-08-09
>
> We are glad the reviewer found the paper well-written and found the experiments to “demonstrate the good performance of the proposed method”. We thank the reviewer for their suggestions for improvement, all of which we have accepted and incorporated into the main text:
>
> - We now provide a derivation of the Err formulas on lines 140–141 in the appendix.
> - We provide a precise description of the exact PWKQ method we use (lines 226–227).
> - We have moved the proof in section 3.4 to the appendix.
> - We have added an application experiment, demonstrating the utility of RPCholesky kernel quadrature for a large-scale data analysis problem.
>
> Taken as a whole and with the support of the new experiments (Fig. 1 in the attached PDF) suggested by the reviewer, our numerical results show that RPCholesky is consistently one of the most accurate kernel quadrature methods. RPCholesky kernel quadrature can be orders of magnitude faster than existing state-of-the-art spectrally accurate kernel quadrature schemes, and these benefits persist to a high (1,500)-dimensional chemistry data analysis task. We believe that, with the addition of the reviewer’s suggested application example, our paper makes a convincing demonstration that RPCholesky kernel quadrature represents a meaningful contribution to the kernel quadrature field.

---

> > ### Comment · Reviewer_yiWH · 2023-08-21
> >
> > Thank you for the response. I will increase my score.

---

### Official Review · Reviewer_4xjP · 2023-07-28

**Soundness:** 4 excellent
**Presentation:** 3 good
**Contribution:** 3 good
**Rating:** 7
**Confidence:** 2

**Summary:**

The recently proposed Randomly Pivoted Cholesky procedure is an efficient and accurate method for approximating symmetric PSD (e.g. kernel) matrices. It builds a Nystrom approximation via randomizing the choice of pivot in the classical Cholesky decomposition procedure.
The authors of the present submission transport the same sampling idea to the continuous case and develop new kernel quadrature rules.
The proposed algorithm is based on rejection sampling, it's simple to implement and it's much faster than many other kernel quadrature procedures. Rigorous theoretical guarantees and experimental analysis with synthetic data accompany the algorithm.

**Strengths:**

1) Solid theory. The error of the quadrature depends on the eigenvalue distribution of the kernel (it's spectrally accurate).
2) The rejection sampling algorithm is practical to implement and it's efficient.
3) Empirical evaluation supports the claims.

**Weaknesses:**

1) As the authors themselves point out in Section 6 all spectrally accurate quadratures, including this, currently require the exact computation of the Hilbert–Schmidt integral of function $g$, which may limit the applicability of the method. E.g. the experiments are all based on $g\equiv 1$.
2) Although the first sentence of the introduction in lines 12-13 refers to several applications all the experiments in Figures 1 and 2 rely on computing the worst case error of kernel integrals in low dimensions on a given domain with a particular (complicated) kernel. Albeit this style of synthetic evaluation seems common in the cited prior work I believe it would increase the impact of the paper if the authors demonstrated end to end improvements on an actual machine learning tasks that requires computing these integrals. Including a concrete application would boost the accessibility and motivation for non experts reading the paper as well in my opinion.


**Questions:**

1) The analysis and experiment focus on and minimize the worst case error (line 140). What, if anything, would change if you considered some expected error (loss) when $f$ is drawn from some distribution?
2) Since the experiments in Figure 2) are rather low dimensional ($d=1$ and $d=3$) could you also include a Quasi-Monte Carlo baseline with equal and optimal (eq (5)) weights similarly to current Monte Carlo and IID? Since QMC is not spectrally accurate one would still expect RPCholesky to have much lower error, the gap might shrink though. Could you also consider running experiments with much higher $d$?


**Limitations:**

Theory paper, societal impact is not applicable. The authors do a thorough job in disclosing and discussing technical limitations.

---

> ### Author Rebuttal · Authors · 2023-08-09
>
> Thanks to the reviewer for their considered remarks. We are very glad to hear the reviewer found the theory and algorithms of the paper to be solid and practical. Below, we address the reviewer’s questions and suggestions:
>
> ## Application
>
> At the suggestion of this review and others, we have included a new experiment to the paper (Fig. 1) featuring a concrete application with high (1,500)-dimensional chemistry data. We agree with the reviewer that including this example will help make the paper more interesting and accessible to a wider audience. We thank the referee for this suggestion.
>
> ## Choice of Loss Function
>
> The reviewer asks what would change if considered the expected loss over a random function $f$. Fortunately, there is a quite satisfactory answer. Let $f$ be a Gaussian process with covariance $k$ and consider the loss$$\tilde{\mathrm{Err}}^2(\mathsf{S},\boldsymbol{w};g) \coloneqq \mathbb{E} \left( \int f(x) g(x) \, \mathrm{d}\mu - \sum_{i=1}^n w_i f(s_i) \right)^2.$$Then $\tilde{\mathrm{Err}}^2(\mathsf{S},\boldsymbol{w};g) = \mathrm{Err}^2(\mathsf{S},\boldsymbol{w};g)$ is exactly the same as the worst-case loss defined in the paper. Here is a brief sketch of the proof. The Gaussian process $f$ has a Karhunen–Loève expansion $f = \sum_i Z_i (\sqrt{\lambda_i} e_i)$ in terms of the $\langle\cdot,\cdot\rangle_{\mathcal{H}}$-orthonormal basis $(\sqrt{\lambda_i} e_i)$, where the $Z_i$ are iid standard normal. Writing $\int f(x) g(x) \, \mathrm{d}\mu - \sum_{i=1}^n w_i f(s_i) = \langle f, Tg - \sum_{i=}^n w_i k(\cdot,s_i) \rangle_{\mathcal{H}} \eqqcolon \langle f, \ell \rangle_{\mathcal{H}}$, we can expand $\ell = Tg - \sum_{i=1}^n w_i k(\cdot,s_i) = \sum_j a_j (\sqrt{\lambda_j} e_j)$ in this orthonormal basis as well. Thus, by orthonormality of  $(\sqrt{\lambda_i} e_i)$,$$\tilde{\mathrm{Err}}^2(\mathsf{S},\boldsymbol{w};g) = \mathbb{E}\langle f, \ell\rangle_{\mathcal{H}}^2 = \mathbb{E} \left( \sum_i a_i Z_i \right)^2= \sum_i a_i^2 = \lVert\ell\rVert_{\mathcal{H}}^2 = \mathrm{Err}^2(\mathsf{S},\boldsymbol{w};g).$$
>
> ## Quasi-Monte Carlo
>
> See the attached Fig. 3 (left panel) for the $s = d = 3$ example with the quasi-Monte Carlo baselines included. With the optimal weights, the performance is quite similar to randomly pivoted Cholesky and continuous volume sampling. Our results agree with [19, Fig. 1], which show optimally weighted QMC to be among the best performers for their 2D and 3D experiments.
>
> To concur with one of the referee’s other remarks, we agree that these low-dimensional Sobolev examples are an odd choice to have become such standard benchmarks in the kernel quadrature literature, as existing quadrature methods (optimally weighted QMC, Gaussian, Clenshaw–Curtis, etc.) are already quite effective for these low-dimensional examples. Quasi-Monte Carlo integration is well-known to become less effective for dimensions larger than, say, six to ten (Morokoff and Caflisch), and would thus have no hope of addressing the 1,500-dimensional chemistry application in Fig. 1.
>
> Morokoff, W. J., & Caflisch, R. E. (1995). Quasi-Monte Carlo Integration. Journal of Computational Physics, 122(2), 218–230. https://doi.org/10.1006/jcph.1995.1209

---

> > ### Comment · Reviewer_4xjP · 2023-08-21
> >
> > Thanks so much for the detailed reply and for adding the experiment with the chemistry task. I believe all these, especially the latter, strengthen the submission further.

---

### Official Review · Reviewer_dsgV · 2023-07-28

**Soundness:** 3 good
**Presentation:** 4 excellent
**Contribution:** 2 fair
**Rating:** 4
**Confidence:** 5

**Summary:**

This article studies a family of quadrature rules suited for functions that belongs to an RKHS. The proposed construction is based on an algorithm called randomly pivoted Cholesky (RPCholesky). The latter was studied for the task of approximating positive-semidefinite matrices. In particular, the authors extended RPC to continuous domain, and investigated its effectiveness for kernel quadrature. More specifically, the authors studied the quadrature rule obtained by taking the nodes to be the configuration of points that RPCholesky and by taking the weights that minimises the worst-case integration error on the unit ball of the RKHS (WCE). In particular, they proved that the expected value of the squared WCE of this quadrature rule converges to 0, with a rate of convergence that depends on the eigenvalues of the integration operator. Moreover, they studied the expected runtime of the proposed algorithmes.

The contributions of the article may be summarised as follows:
* An extension of the RPCholesky algorithm to a continuous domain (Algorithm 1) along with a detailed implementation of it using rejection sampling (Algorithm 2).
* An implementation of RPCholesky with optimized rejection sampling (Algorithm 4).
* A generic result (Theorem 1) gives the upper bound of the expected squared WCE for the proposed quadrature rule.
* The study of the expected runtime of Algorithm 2 (Theorem 4).



**Strengths:**

- The article is very well written.
- The article expands the application of the randomly pivoted Cholesky algorithm into the continuous domain, establishing another bridge between the numerical linear algebra community and the kernel quadrature community.
- The proof of the main result (Theorem 1) is original and may be beneficial for the community of kernel quadrature in the future.


**Weaknesses:**

At first sight, the proposed algorithm seems to bring a significant algorithmic improvement upon continuous volume sampling. This is at least what we may understand reading the comments of Figure 2 (line 235-line 238). While, the relative improvement is undeniable, RPCholesky suffers from the same limitation of continuous volume sampling that is poor scalability for large values of $N$, especially for smooth kernels. Figure 2 of the article is corroborating this claim: the dependence of the empirical expected time as a function of $N$ seems to be polynomial. This is particularly the case in dimension 1,  where we have $\lambda_m = \mathcal{O}(m^{-2s})$ so that $1/\sum_{m=N+1}^{+\infty}\lambda_{m} = \mathcal{O}(N^{2s+1})$. In other words, the smoother is the kernel the longer it will takes to sample from the RPCholesky, which is exactly the same problem with Gibbs sampling proposed in [1], which provably approximate continuous volume sampling. It turns out that Algorithm 1 is nothing but the algorithm proposed in [1] to initialise an MCMC algorithm that approximate continuous volume sampling; see Algorithm 1 in [1].  The difference between the two works is in the subroutine that consist in sampling from the probability measure $x \mapsto k(x,x) \mathrm{d}\mu(x)/\int_{\mathcal{X}}k(u,u)\mathrm{d}\mu(u)$: [1] proposed to use a ‘stochastic greedy’ algorithm using rejection sampling, while this article proposed to use RPCholesky with rejection sampling. Interestingly, the two algorithms have an expected time that depends on $1/\sum_{m=N+1}^{+\infty}\lambda_{m}$, where is the number of nodes in the quadrature rule, see Lemma 5.1 in [1]. In other words, this work does not break the ‘curse of smoothness’ that hinders the scalability of continuous volume sampling.

[1] Rezaei, A., & Gharan, S. O. (2019, May). A polynomial time MCMC method for sampling from continuous determinantal point processes. In International Conference on Machine Learning (pp. 5438-5447). PMLR.

[2] Anari, N., Gharan, S. O., & Rezaei, A. (2016, June). Monte Carlo Markov chain algorithms for sampling strongly Rayleigh distributions and determinantal point processes. In Conference on Learning Theory (pp. 103-115). PMLR.


**Questions:**

- The empirical comparison in section 5 was limited to Sobolev spaces of order s=3. It may be interesting to compare RPCholesky to other algorithms on other RKHSs.
- The empirical experiments make use of a continuous analog of a ’basis-exchange’ MCMC algorithm proposed in [2] instead of the MCMC algorithm based on Gibbs sampling proposed in [1]. Up to my knowledge, only the latter was studied theoretically, and it is possible that the convergence of ’basis-exchange’ MCMC algorithm is slower than the Gibbs sampling.

---

> ### Author Rebuttal · Authors · 2023-08-09
>
> We appreciate the reviewer's very incisive comments. We are delighted to hear that they found the article well-written and that they enjoyed the further connections our article draws between kernel quadrature and numerical linear algebra. Below, we address your concerns and questions:
>
> ## Curse of Smoothness
>
> We agree that Algorithm 2 does not break the curse of smoothness. However, RPCholesky has a substantially better scaling with $n$ compared to CVS. [1, Thm. 1.3] suggests that $O(n^5 \log n)$ MCMC steps are needed to sample from the CVS process. For kernels with polynomial spectral decay, RPCholesky provably produces near-optimal quadrature nodes with just $n$ steps. **The improvement in runtime over [1] is log-quartic $O(n^4 \log n)$.**
>
> As the reviewer suggests, RPCholesky is identical to the "warm start" used to initialize the CVS MCMC sampler of [1]. We believe the connections between RPCholesky and DPP/CVS to be a feature, not a bug. One way to frame the results of this paper is that the initialization phase of [1] is already enough to get near-optimal quadrature nodes, and the costly $O(n^5 \log n)$ MCMC steps required to sample from the CVS distribution can be avoided.
>
> We also bring the reviewer's attention to Algorithm 4. This algorithm blunts the curse of smoothness by incorporating a small number of global optimization steps to choose a better threshold for rejection sampling. There are certainly limitations to Algorithm 4: We do not have theory for this variant, and the requirement for optimization limits the range of applicability. However, in many settings where the curse of smoothness really starts to bite, we hope Algorithm 4 will be a useful tool in practice (see below for more discussion on this point).
>
> ## Gibbs Sampler vs Basis Exchange
>
> Our paper used the basis exchange sampler rather than the Gibbs sampler because a direct implementation of the Gibbs sampler as described in [1] for our problem would be intractable. Indeed, using a direct implementation of [1, Alg. 3] and running for an optimistic $10n$ MCMC steps (rather than the theoretically sanctioned $O(n^5 \log n)$ steps [1, Thm. 1.3] or the empirically suggested $O(n^2)$ steps [1, Fig. 1]), **the Gibbs sampler required 30 minutes just to produce a single sample of size 20 for the $s=3$, $d=1$ Sobolev space. Algorithm 4 requires just 0.1 seconds to produce an RPCholesky sample of this size.** We hope this example provides significant evidence to the reviewer that, while we have not provably conquered the curse of smoothness, Algorithm 4 can improve the state of affairs significantly in practice.
>
> We can improve the runtime of the Gibbs sampler of [1] by the ideas of Algorithm 4 to sample the conditional 1-DPP subproblems needed to implement the algorithm. Specifically, at each MCMC step, we compute an upper bound on the residual kernel using a global optimization step and use this to determine the optimal "$M$" parameter for rejection sampling. We again run for $10n$ MCMC steps. Results are shown in Fig. 2 in the attached PDF. The quadrature error with CVS sampled by the Gibbs sampler is worse than both RPCholesky and for CVS sampling using basis exchange. **Even with the optimistic choice to run for only $10n$ MCMC steps and the algorithmic improvements adopted from Alg. 4, the Gibbs sampler takes 120$\times$ longer than RPCholesky to sample $n=64$ nodes**
>
> ## Conclusion
>
> We appreciate the reviewer's thoughtful comments, which demonstrate both a close reading of our paper and a lot of knowledge of the related literature. We hope that the above discussion is convincing that RPCholesky results in a substantial acceleration over the existing state-of-the-art spectrally accurate kernel quadrature schemes in both theory and practice.

---

> > ### Comment · Reviewer_dsgV · 2023-08-21
> >
> > Thank you for all the clarifications that you have provided.

---

### Author Rebuttal · Authors · 2023-08-09

We thank the reviewers for their very thoughtful comments. We have revised the paper to address any questions and concerns raised and have accepted all specific recommended changes by all reviewers. We have responded to the questions of each reviewer in the individual responses, and we look forward to addressing any lingering concerns in the discussion period.

Several reviewers asked us to add an additional example, with different reviewers requesting an example with a higher dimension, a different kernel, or an application. In the interest of conciseness, we have addressed these concerns jointly by adding a new section to the paper featuring an application to large chemistry datasets.

The experimental setup is as follows: Following inspiration from [19], we apply the kernel quadrature formalism to a finite dataset, using kernel quadrature to estimate the average value of a chemical property over a subsample of 20,000 molecules from the QM9 dataset. Each molecule is represented as a vector in **1,500-dimensional space** using the many-body tensor representation, a standard featurization in chemical machine learning. We use a Gaussian kernel; with this new application, our paper now includes the Matérn, Sobolev, and Gaussian kernels. We show both the worst-case error and, following the suggestion of 4xjP, the error for a specific chemical property, the isotropic polarizability. We omit some of the methods from the previous comparison because of their high computational cost, and add in the greedy Nyström method (Fine and Scheinberg) with optimal weights (8).

Fig. 1 in the attached PDF shows the results. RPCholesky improves over all other methods we tested by a factor of three for isotropic polarizability and has the lowest worst-case error.

We thank the reviewers for the suggestion to include such an example. We hope that this example is both simple enough to be widely appreciated, and demonstrates the flexibility and utility of RPCholesky kernel quadrature for high-dimensional data problems.

Fine, S., & Scheinberg, K. (2001). Efficient SVM training using low-rank kernel representations. Journal of Machine Learning Research, 2, 243–264.

---

### Decision · Program_Chairs · 2023-09-21

**Decision:**

Accept (spotlight)

**Comment:**

This is a high-quality submission that makes a major contribution to kernel cubature and is presented excellently.  I am delighted to recommend its acceptance at NeurIPS!